# Recursive Construction of Stable Assemblies of Recurrent Neural Networks

## Abstract

Advanced applications of modern machine learning will likely involve combinations of trained networks, as are already used in spectacular systems such as DeepMind's AlphaGo. Recursively building such combinations in an effective and stable fashion while also allowing for continual refinement of the individual networks - as nature does for biological networks - will require new analysis tools. This paper takes a step in this direction by establishing contraction properties of broad classes of nonlinear recurrent networks and neural ODEs, and showing how these quantified properties allow in turn to recursively construct stable networks of networks in a systematic fashion. The results can also be used to stably combine recurrent networks and physical systems with quantified contraction properties. Similarly, they may be applied to modular computational models of cognition. We perform experiments with these combined networks on benchmark sequential tasks (e.g. permuted sequential MNIST) to demonstrate their capacity for processing information across a long timescale in a provably stable manner.

## 1 Introduction

Neuro-inspired machine learning has profoundly altered many fields such as computer vision, natural language processing, and computational neuroscience (Bengio et al., 2017; Hassabis et al., 2017). While models trained with e.g. deep learning are remarkably powerful, they are for the most part 'black boxes'. This opaqueness can be dangerous in safety-critical applications, such as autonomous driving or human-centered robotics, and it limits conceptual progress. In the case of recurrent models, one difficulty is that providing a certificate of *stability* is currently impossible or computationally impractical. Given that stability is a fundamental property of dynamical systems – and is intimately linked to concepts of control, generalization, data-efficiency, and robustness – being able to guarantee the stability of a recurrent model is an important step towards making sure deep models behave as we expect them to.

In this spirit, there has been a recent flux of work focusing on applications of contraction analysis (Lohmiller & Slotine, 1998) to recurrent models. Loosely speaking, a dynamical system is said to be contracting if any two of its trajectories converge to each other exponentially, regardless of initial conditions. A primary advantage of contraction analysis is that it is directly applicable to non-autonomous systems, which the vast majority of recurrent models are, allowing in turn modular contraction-preserving combination properties to be derived (Slotine & Lohmiller, 2001; Slotine, 2003). We include a mathematical primer on contraction analysis in Section A1.

Within this line of research, our paper has two main aims: 1) To provide simple contraction conditions for *continuous-time* recurrent neural networks and 2) To show how these continuous-time contraction conditions imply a combination property. Using both aims, we proceed to implement stable combination networks that exhibit state-of-the-art (SOTA) performance on multiple sequential image classification tasks with a small number of trainable parameters.

### 1.1 Previous Work on RNN Stability

To briefly review the current literature on application of contraction analysis to recurrent models, we first note that the 'Echo-State Condition' introduced in Jaeger (2001) is equivalent to discrete-time contraction in the identity metric. A later generalization of this condition included a diagonal metric

(Buehner & Young, 2006). In the context of neuroscience, contraction analysis has been applied to analyzing the dynamics of winner-take-all networks (Rutishauser et al., 2011; 2015) as well as networks with synaptic plasticity (Kozachkov et al., 2020). In the machine learning context, Miller and Hardt recently rederived an 'echo-state property' for discrete recurrent models, and went on to prove that these contracting recurrent models could be well-approximated by feedforward networks in certain cases (Miller & Hardt, 2018). More recently still, in a series of papers Revay, Wang, and Manchester applied contraction analysis to discrete-time recurrent networks (Revay & Manchester, 2020; Revay et al., 2021; 2020a), expanding the class of models considered in Miller & Hardt (2018).

In addition to contraction (which amounts to a strong form of non-autonomous exponential stability) there has been a considerable amount of work attempting to enforce weaker forms of stability in RNNs, such as *autonomous* stability (Erichson et al., 2021; Chang et al., 2019). Unfortunately, as we discuss in Section 2.3, autonomous stability does not in general imply non-autonomous stability, so it is not clear what stability properties these RNNs possess when driven with external input.

There is also a line of work which uses orthogonal weight matrices to avoid the vanishing/exploding gradient problem during training. Orthogonality is typically ensured during training via a parameterization (Arjovsky et al., 2016; Lezcano-Casado & Martınez-Rubio, 2019) – for example, by exploiting the fact that the matrix exponential of a skew-symmetric matrix is orthogonal, as is done in Lezcano-Casado & Martınez-Rubio (2019). These works focus on parameterizations of *individual* RNNs which ensure stability is preserved during training. By contrast, our work focuses on 'combined' RNNs (i.e network of networks) and provides a parameterization on the *connections between stable subnetworks* such that training preserves the overall network stability (Figure 1).

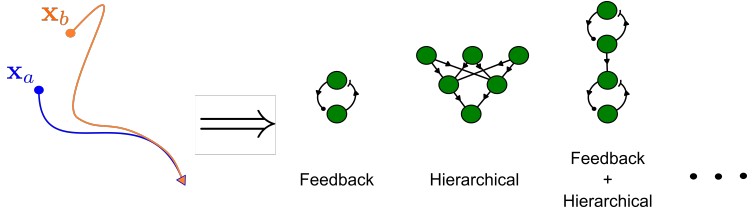

Figure 1: Our stability certificate implies a modularity principle. It may be used to recursively construct complicated 'networks of networks' while automatically maintaining stability.

## 1.2 COMBINATION NETWORKS

The combination and reuse of primitive "modules" has enabled a great deal of progress in computer science, and is also a key theme in biological evolution, particularly apparent in cortical structure of the human brain. In fact, it is thought that the majority of traits that have developed over the last 400+ million years are the result of evolutionary forces acting on regulatory elements that combine core components, rather than mutations to the core components themselves. This mechanism of action makes meaningful variation in population phenotypes much more feasible to achieve, and is appropriately titled "facilitated variation" (Gerhart & Kirschner, 2007). In addition to the biological evidence for facilitated variation, computational models have demonstrated that this approach produces populations that are better able to generalize to new environments (Parter et al., 2008), an ability that will be critical to further develop in deep learning systems.

While the benefits of building modular systems are clear (Simon, 1962), as in DeepMind's AlphaGo for example (Silver et al., 2016), there is no general guarantee that a combination of stable systems will itself be stable. Thus the tractability of these evolutionary processes hinges on some mechanism for ensuring stability of combinations. Because contraction analysis tools allow complicated contracting systems to be built up recursively from simpler elements, this form of stability is well suited for biological systems (Slotine & Lohmiller, 2001; Slotine & Liu, 2012). Note also that contracting combinations can be made between systems with very different dynamics, as long as those dynamics are contracting. Here, we describe two common forms of contracting system combinations – hierarchical and feedback – that automatically guarantee overall system stability (Figure 1).

Ultimately, cognitive models are moving increasingly toward study of multi-area dynamics, but many questions remain on how to best train and evaluate such networks (Yang et al., 2018; Yang &

Molano-Mazon, 2021). Understanding how different brain regions interact harmoniously to produce a unified percept/action will require new ideas and analysis tools.

## 2 MATHEMATICAL RESULTS

### 2.1 CONTRACTION CONDITIONS FOR INDIVIDUAL, CONTINUOUS-TIME RNNS

We consider the following continuous-time RNN, evolving according to the following equations:

$$\tau\dot{\mathbf{x}} = -\mathbf{x} + \mathbf{W}\phi(\mathbf{x}) + \mathbf{u}(t) \tag{1}$$

where $\mathbf{x} \in \mathbb{R}^n$, $\phi$ is a static nonlinearity such that $0 \leq \phi' \leq g$, $\mathbf{u}(t)$ is some input (potentially time-varying) and $\tau > 0$. We do not constrain the sign of $\phi$. Example nonlinearities $\phi(x)$ that satisfy the constraints are $\tanh(ax)$, $\log(1 + e^x)$, and $\max(0, x)$, with $g = a, 1, 1$ respectively. Note that this class of RNNs is equivalent to another commonly used class of RNNs where the terms $\mathbf{W}\mathbf{x} + \mathbf{u}$ appears inside $\phi(\cdot)$. See Section A2 or (Miller & Fumarola, 2012) for details. Our mathematical results therefore apply equally well to both types of RNNs.

We seek a stability certificate for this system in terms of the recurrent weight matrix $\mathbf{W}$. We are specifically interested in restricting $\mathbf{W}$ such that the RNN is globally *contracting*. We do this with the goal of recursively combining these contracting RNNs with other contracting RNNs to make large, complicated, stable 'networks of networks'. Here we derive five restrictions on $\mathbf{W}$ which ensure contraction for the continuous-time RNN defined by equation 1. To the best of our knowledge these contraction conditions are novel. All proofs can be found in the supplemental Section A4.

**Theorem 1** (Absolute Value Restricted Weights). *Let $|\mathbf{W}|$ denote the matrix formed by taking the element-wise absolute value of $\mathbf{W}$. If there exists a positive, diagonal $\mathbf{P}$ such that:*

$$\mathbf{P}(g|\mathbf{W}| - \mathbf{I}) + (g|\mathbf{W}| - \mathbf{I})^T\mathbf{P} \prec 0$$

*then equation 1 is contracting in metric $\mathbf{P}$. Moreover, if $W_{ii} \leq 0$, then $|W|_{ii}$ may be set to zero to reduce conservatism.*

Note that when $g = 1$, Theorem 1 can be checked by checking for linear stability of $|\mathbf{W}| - \mathbf{I}$.

**Theorem 2** (Symmetric Weights). *If $\mathbf{W} = \mathbf{W}^T$ and $g\mathbf{W} \prec \mathbf{I}$, then (1) is contracting.*

**Theorem 3** (Product of Diagonal and Orthogonal Weights). *If there exists positive diagonal matrices $\mathbf{P}_1$ and $\mathbf{P}_2$, as well as $\mathbf{Q} = \mathbf{Q}^T \succ 0$ such that*

$$\mathbf{W} = -\mathbf{P}_1\mathbf{Q}\mathbf{P}_2$$

*then (1) is contracting in metric $\mathbf{M} = (\mathbf{P}_1\mathbf{Q}\mathbf{P}_1)^{-1}$.*

**Theorem 4** (Triangular Weights). *If $g\mathbf{W} - \mathbf{I}$ is triangular and Hurwitz, then (1) is contracting in a diagonal metric.*

**Theorem 5** (Singular Value Restricted Weights). *If there exists a positive diagonal matrix $\mathbf{P}$ such that:*

$$g^2\mathbf{W}^T\mathbf{P}\mathbf{W} - \mathbf{P} \prec 0$$

*then (1) is contracting in metric $\mathbf{P}$.*

### 2.2 THE MODEL: NETWORK OF NETWORKS

The RNN in equation 1 is a *subnetwork* of our model. Our goal is to combine these subnetworks into a 'network of networks' in a manner that preserves the stability of the underlying modules. In particular, we seek to parameterize and learn the inter-module connections. To do this, we prove and make extensive use of the following theorem:

**Theorem 6** (Network of Networks). *Consider a collection of $p$ subnetwork RNNs governed by equation 1. Assume that these RNNs each have hidden-to-hidden weight matrices $\{\mathbf{W}_1, \ldots, \mathbf{W}_p\}$ and are independently contracting in metrics $\{\mathbf{M}_1, \ldots, \mathbf{M}_p\}$. Define the block matrices $\tilde{\mathbf{W}} \equiv BlockDiag(\mathbf{W}_1, \ldots, \mathbf{W}_p)$ and $\tilde{\mathbf{M}} \equiv BlockDiag(\mathbf{M}_1, \ldots, \mathbf{M}_p)$, as well as the overall state vector $\tilde{\mathbf{x}}^T \equiv (\mathbf{x}_1^T \cdots \mathbf{x}_2^T)$. Then the following 'network of networks' is globally contracting in metric $\tilde{\mathbf{M}}$:*

$$\tau\dot{\tilde{\mathbf{x}}} = -\tilde{\mathbf{x}} + \tilde{\mathbf{W}}\phi(\tilde{\mathbf{x}}) + \mathbf{u}(t) + \mathbf{L}\tilde{\mathbf{x}}$$
$$\mathbf{L} \equiv \mathbf{B} - \tilde{\mathbf{M}}^{-1}\mathbf{B}^T\tilde{\mathbf{M}} \tag{2}$$

*Where $\mathbf{B}$ is an arbitrary square matrix. Note that we are agnostic to the particular contraction condition here – the subnetwork RNNs can satisfy any of the theorems in the preceding section.*

### 2.3 ON STABILITY THEOREMS FOR RNNS

Several recent papers in machine learning, e.g (Haber & Ruthotto, 2017; Chang et al., 2019), claim that a sufficient condition for stability of the nonlinear system:

$$\dot{\mathbf{x}} = \mathbf{f}(\mathbf{x}, t)$$

is that the associated Jacobian matrix $\mathbf{J}(\mathbf{x}, t) = \frac{\partial \mathbf{f}}{\partial \mathbf{x}}$ has eigenvalues whose real parts are strictly negative, i.e:

$$\max_i \text{Re}(\lambda_i(\mathbf{J}(\mathbf{x}, t))) \leq -\alpha$$

with $\alpha > 0$. This claim is generally false. For a counter-example, see Section 4.4.2 in (Slotine & Li, 1991).

However, in the *specific* case of the RNN (1), it appears that the eigenvalues of the symmetric part of $\mathbf{W}$ do provide information on global stability in a number of applications. For example, in (Matsuoka, 1992) it was shown that if $\mathbf{W}_s = \frac{1}{2}(\mathbf{W} + \mathbf{W}^T)$ has all its eigenvalues less than unity, and $\mathbf{u}$ is constant, then (1) has a unique fixed point that is globally asymptotically stable. It is easy to see that this condition also implies that the real parts of the eigenvalues of the Jacobian are uniformly negative. Moreover, in (Chang et al., 2019) it was shown that setting the symmetric part of $\mathbf{W}_s = \frac{1}{2}(\mathbf{W} + \mathbf{W}^T)$ almost equal to zero (yet slightly negative) led to rotational, yet stable dynamics in practice. This leads us to the following theorem, which shows that if the slopes of the activation functions change sufficiently slowly as a function of time, then the condition in (Matsuoka, 1992) in fact implies global contraction of (1).

**Theorem 7.** *Let $\mathbf{D}$ be a positive, diagonal matrix with $D_{ii} = \frac{d\phi_i}{dx_i}$, and let $\mathbf{P}$ be an arbitrary, positive diagonal matrix. If:*

$$(g\mathbf{W} - \mathbf{I})\mathbf{P} + \mathbf{P}(g\mathbf{W}^T - \mathbf{I}) \preceq -c\mathbf{P}$$

*and*

$$\dot{\mathbf{D}} - cg^{-1}\mathbf{D} \preceq -\beta\mathbf{D}$$

*for $c, \beta > 0$, then (1) is contracting in metric $\mathbf{D}$ with rate $\beta$.*

It has been conjectured that diagonal stability of $g\mathbf{W} - \mathbf{I}$ is a sufficient condition for global contraction of the RNN 1 (Revay et al., 2020b), however this has been difficult to prove. To better characterize this conjecture, we present Theorem 8, which shows by way of counterexample that diagonal stability of $g\mathbf{W} - \mathbf{I}$ does NOT imply global contraction in a *constant* metric for (1).

**Theorem 8.** *Satisfaction of the condition $\quad g\mathbf{W}_{sym} - \mathbf{I} \prec 0 \quad$ is **NOT** sufficient to show global contraction of the general nonlinear RNN (1) in any constant metric. High levels of antisymmetry in $\mathbf{W}$ can make it impossible to find such a metric, which we demonstrate via a $2 \times 2$ counterexample of the following form, with $c \geq 2$: $\mathbf{W} = \begin{bmatrix} 0 & -c \\ c & 0 \end{bmatrix}$*

## 3 EXPERIMENTS

It is possible that imposing a strong stability constraint on the RNNs precludes them from performing well on tasks that require information processing over long timescales. To investigate whether or not this occurs with our stable combination networks, we trained a variety of provably stable RNNs on benchmark sequential image classification tasks - sequential MNIST, permuted sequential MNIST, and sequential CIFAR10 - as these are frequently used to measure an RNN's ability to use information from across a long sequence (Le et al., 2015; Chang et al., 2019). For all the tasks, images are presented pixel-by-pixel, and the network makes a prediction at the end of the sequence. In the case of permuted seqMNIST, the pixels are presented in a fixed but random order.

Recall that our proposal is to constrain the stability of subnetworks in a way that makes training the combined network (using Theorem 6) stability-preserving. The particular choice of subnetwork constraint influences the resulting training process and performance. We selected two stability constraints from our results above to experiment with: Theorems 1 and 5. In the case of Theorem 1, we did not train the individual subnetworks' hidden-to-hidden weight matrices, only training the connections *between* subnetworks (Figure 2A). In the case of Theorem 5 we trained all the parameters of the model (Figure 2B).

We refer to networks with subnetworks constrained by Theorem 1 as 'Sparse Combo Nets' (because, as we will detail below, sparse hidden-to-hidden weights more readily satisfy Theorem 1). Similarly, we refer to networks with subnetworks constrained by Theorem 5 as 'SVD Combo Nets'. Throughout the experimental results we refer to combination networks of different sizes using the notation '$p \times n$ network'. Such a network consists of $p$ distinct subnetwork RNNs, with each such subnetwork RNN containing $n$ units. For all subnetworks we use the ReLU activation function.

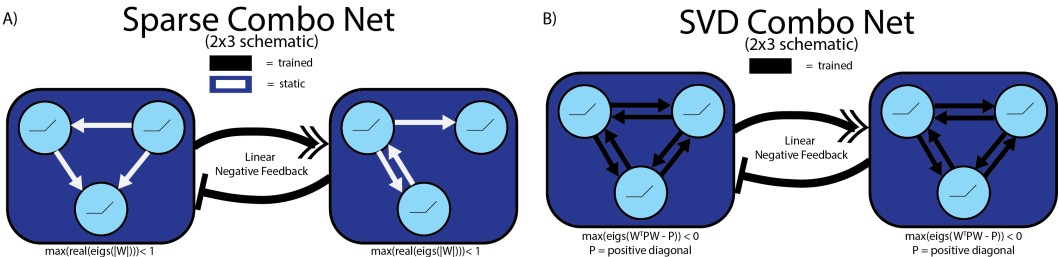

Figure 2: Summary of network architectures. Nonlinear subnetworks are constrained to meet either Theorem 1 via sparse initialization (A) or Theorem 5 via direct parameterization (B). Linear negative feedback connections are trained between the subnetworks according to 6. See Section A5 for additional details on model definitions.

Our networks performed better than or comparably to other stable RNNs across all tasks, especially when leveraging sparsity in the subnetworks. Our best test accuracies of $96.94\%$ on permuted seqMNIST and $64.75\%$ on seqCIFAR10 not only set a new SOTA for stability-guaranteed RNNs, but are also competitive with the overall SOTA while having many fewer parameters (Table 1).

### 3.1 NETWORK INITIALIZATION AND TRAINING DETAILS

For the Sparse Combo Net we were not able to find a direct parameterization to use during training, but it is straightforward to randomly generate matrices with a particular likelihood of meeting the condition by selecting an appropriate density level and limit on entry magnitude. As every RNN meeting this contraction condition has a well-defined stable LTI system contracting in the same metric, it is not only easy to verify, but also easy to find a contraction metric to use in our training algorithm (Theorem 6). Solving for $\mathbf{M}$ in $-\mathbf{I} = \mathbf{M}\mathbf{A} + \mathbf{A}^T\mathbf{M}$ will produce a valid metric for any stable LTI system $\mathbf{A}$ (Slotine & Li, 1991). We therefore randomly generate fixed subnetworks satisfying Theorem 1 and train only the linear connections between them (Figure 3), as well as the linear input and output layers.

Figure 3: Example $3 \times 16$ Sparse Combo Net. The hidden-to-hidden nonlinear weights ($\tilde{\mathbf{W}}$ in Theorem 6) are initialized based on a set sparsity, and do not changed over training (A). The linear inter-subnetwork connections ($\mathbf{L}$ in Theorem 6) are constrained to be antisymmetric (with respect to the overall network metric), and are updated in the training process (B).

For the SVD Combo Net, we ensured contraction by directly parameterizing each of the $\mathbf{W}_i$ ($i = 1, 2 \ldots p$) as:

$$\mathbf{W}_i = \mathbf{\Phi}_i^{-1} \mathbf{U}_i \mathbf{\Sigma}_i \mathbf{V}_i^T \mathbf{\Phi}_i \tag{3}$$

where $\mathbf{\Phi}_i$ is diagonal and nonsingular, $\mathbf{U}_i$ and $\mathbf{V}_i$ are orthogonal, and $\mathbf{\Sigma}_i$ is diagonal with $\Sigma_{ii} \in [0, g^{-1})$. We ensure orthogonality of $\mathbf{U}_i$ and $\mathbf{V}_i$ during training by exploiting the fact that the matrix exponential of a skew-symmetric matrix is orthogonal, as was done in (Lezcano-Casado & Martınez-Rubio, 2019). The network constructed from these subnetworks using Theorem 6 is contracting in metric $\tilde{\mathbf{M}} = \text{BlockDiag}(\mathbf{\Phi}_1^2, \ldots, \mathbf{\Phi}_p^2)$.

For both networks, we constrained the matrix $\mathbf{B}$ in Theorem 6 to reflect underlying modularity assumptions. In particular, we only train the off-diagonal blocks of $\mathbf{B}$ and mask the diagonal blocks. We do this to maintain the interpretation of $\mathbf{L}$ as the matrix containing the connection weights *between* different modules, as diagonal blocks would correspond to self-connections. Furthermore, we only train the lower-triangular blocks of $\mathbf{B}$ while masking the others, to increase training speed.

Unless specified otherwise, all networks were trained for 150 epochs, using an Adam optimizer with initial learning rate 1e-3 and weight decay 1e-5. The learning rate was cut to 1e-4 after 90 epochs and to 1e-5 after 140. After identifying the most promising settings, we ran repetitions trials on the best networks for 200 epochs with learning rate cuts after epochs 140 and 190. Detailed information on network initialization and hyperparameter tuning is provided in Section A3.

## 3.2 RESULTS

The Sparse Combo Net architecture achieved the highest overall performance on permuted seqM-NIST and seqCIFAR10 (96.94% and 64.75% best performance, respectively). On both these tasks, the Sparse Combo Net achieved SOTA for stable RNNs (Table 1). Furthermore, we were able to reproduce these scores over several repetitions (Section 3.2.3). Along with repeatability of results, we also show that the contraction constraint on the connections between subnetworks ($\mathbf{L}$ in Theorem 6) is important for performance, particularly in the Sparse Combo Net.

Additionally, we profile how various architecture settings impact performance of our networks. We found that the sparsity of the hidden-to-hidden weights in Sparse Combo Net had a large impact on the final network performance (Section 3.2.1). In both networks, we found that increasing the total number of neurons improved task performance, but with diminishing returns (Section 3.2.2).

### 3.2.1 EXPERIMENTS WITH SPARSITY

Because of the link between sparsity and stability as well as the biological relevance of sparsity, we explored in more detail how subnetwork sparsity affects the performance of Sparse Combo Net. We ran a number of additional experiments on the permuted seqMNIST task, varying sparsity level while holding network size and other hyperparameters constant. Here we use "$n\%$ sparsity level" to refer to a network with subnetworks that have just $n\%$ of their weights non-zero.

We observed a large ($> 5$ percentage point) performance boost when switching from a 26.5% sparsity level to a 10% sparsity level in the $11 \times 32$ Sparse Combo Net (Figure 4), and subsequently

| Name | Stable RNN? | Params | Seq MNIST Best | PerSeq MNIST Best | Seq CIFAR Best | sMNIST Repeats Mean (n) [Min] | psMNIST Repeats Mean (n) [Min] | sCIFAR10 Repeats Mean (n) [Min] |
|---|---|---|---|---|---|---|---|---|
| LSTM (Chang et al., 2019) | | 68K | 97.3% | 92.7% | 59.7% | — | — | — |
| Transformer (Trinh et al., 2018) | | 0.5M | 98.9% | 97.9% | 62.2% | — | — | — |
| Antisymmetric (Chang et al., 2019) | ? | 36K | 98% | 95.8% | 58.7% | — | — | — |
| Sparse Combo Net | ✓ | 130K | 99.04% | **96.94%** | **64.75%** | — | 96.85% (4) [96.65%] | 64.37% (3) [64.04] |
| Lipschitz (Erichson et al., 2021) | ✓ | 134K | **99.4%** | 96.3% | 64.2% | 99.2% (10) [99.0%] | 95.9% (10) [95.6%] | — |
| Dense IndRNN (Li et al., 2020) | | 83K | **99.48%** | 97.2% | — | — | — | — |
| CKConv (Romero et al., 2021) | | 1M | 99.32% | **98.54%** | 63.74% | — | — | — |
| Trellis (Bai et al., 2019) | | 8M | 99.2% | 98.13% | **73.42%** | — | — | — |

Table 1: Published benchmarks for sequential MNIST, permuted MNIST, and sequential CIFAR10 best test accuracy. Networks are grouped into three categories: baselines, best performing RNNs with claimed stability guarantee*, and networks achieving the current SOTA on each task. Within each grouping, networks are ordered by number of trainable parameters (for CIFAR10 if it differed across tasks). Our network is highlighted. Where possible, we include information on repeatability. *For more on stability guarantees in machine learning, see Section 2.3.

decided to test significantly sparser subnetworks in a $16 \times 32$ Sparse Combo Net. We trained networks with sparsity levels of $5\%$, $3.3\%$, and $1\%$, as well as $10\%$ for baseline comparison (Figure S3A). A $3.3\%$ sparsity level produced the best results, leading to our SOTA performance on permuted seqMNIST. For a component RNN size of just 32 units, this sparsity level is quite small, with only one or two directional connections per neuron on average (Figure S1).

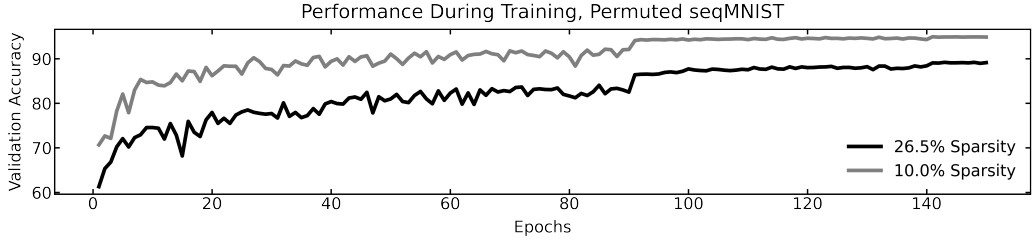

Figure 4: Permuted seqMNIST performance over the course of training for two $11 \times 32$ Sparse Combo Nets with different sparsity levels.

The $16 \times 32$ Sparse Combo Net with $3.3\%$ density also performed SOTA for stable RNNs on sequential CIFAR10. Not only was the test accuracy the highest to date for a provably stable RNN, it was higher than the 1 million parameter CKConv network, which holds the record for permuted seqMNIST accuracy (Table 1). Our network has 130 thousand trainable parameters by comparison.

As sparsity had such a positive effect on task performance, we did additional analyses to better understand why. We found that decreasing the magnitude of non-zero elements while holding sparsity level constant decreased task performance (Figure S3B), suggesting that the effect is driven in part by the fact that sparsity enables higher magnitude non-zero elements while still maintaining stability.

It is also worth noting that upon investigation of the subnetwork weight matrices across these trials, the sparser networks had substantially lower maximum eigenvalue of $|\mathbf{W}|$, suggesting that stronger stability can actually correlate with improved performance on sequential tasks. This could be due to a mechanism such as that described in (Radhakrishnana et al., 2020).

### 3.2.2 Experiments with Network Size

Understanding the effect of size on network performance is important to practical application of these architectures. For both Sparse Combo Net and SVD Combo Net, increasing the number of subnetworks while holding other settings constant (including fixing the size of each subnetwork at 32 units) was able to increase network test accuracy on permuted seqMNIST to a point (Figure 5).

The greatest performance jump happened when increasing from one module (37.1% Sparse Combo Net, 61.8% SVD Combo Net) to two modules (89.1% Sparse Combo Net, 92.9% SVD Combo Net). After that the performance increased steadily with number of modules until saturating at $\sim 97\%$ for Sparse Combo Net and $\sim 95\%$ for SVD Combo Net. Because the internal subnetwork weights are not trained in Sparse Combo Net, it is unsuprising that its performance was substantially worse at the smallest sizes. However Sparse Combo Net surpasses SVD Combo Net by the $12 \times 32$ network size, which contains a modest 384 total units.

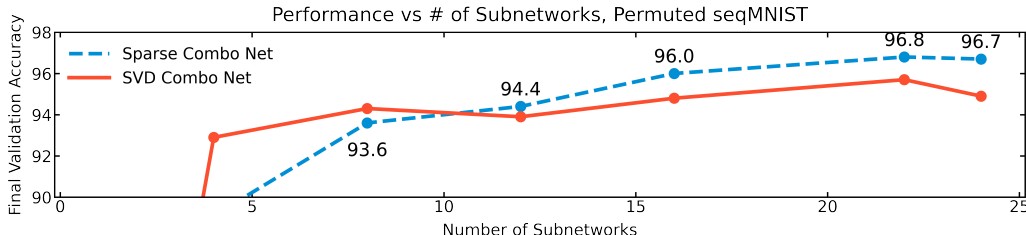

Figure 5: Permuted seqMNIST performance plotted against the number of subnetworks. Each subnetwork has 32 neurons. Results are shown for both Sparse Combo Net and SVD Combo Net.

For the Sparse Combo Net specifically we did additional experiments, both replicating the observed effect of network size using subnetworks with 16 units each (Figure S2A), and evaluating how task performance varies with modularity of a network fixed to have 352 total units (Figure S2B). In the modularity experiment we observed an inverse U shape, with poor performance of a $1 \times 352$ net and an $88 \times 4$ net, and best performance from a $44 \times 8$ net. Note that this experiment compared similar sparsity levels across the different subnetwork sizes. In practice we can achieve better performance with larger subnetworks by leveraging sparsity in a way not possible in smaller subnetworks.

The use of sparsity in subnetworks to improve performance suggests another interesting direction that could enable better scalability of total network size - enforcing sparsity in the linear feedback weight matrix ($\mathbf{L}$). We performed pilot testing of this idea in a $24 \times 32$ Sparse Combo Net, varying the number of feedback connections that were fixed at 0 while all other settings remained as they were in our prior experiments (Table S7). We obtained 65.14% test accuracy on sequential CIFAR10 using a network with only 50% of the possible feedback connections non-zero, surpassing our previous best and using a smaller number of epochs to do so.

### 3.2.3 Repeatability and Controls

Because the Sparse Combo Net does not have the connections within its subnetworks trained, network performance could be particularly susceptible to random initialization. Thus we ran repeatability studies on both permuted sequential MNIST and sequential CIFAR10 using our best network settings and an extened training period. Across four permuted seqMNIST trials with the same network

settings ($16 \times 32$ with subnetwork sparsity level of $3.3\%$), best test accuracy always fell between $96.65\%$ and $96.94\%$, a range much smaller than the differences seen with changing sparsity settings and network size. Three of the four trials showed best test accuracy $\geq 96.88\%$, despite some variability in early training performance (Figure S4). Similarly, across three seqCIFAR10 trials with the same network settings, best test accuracy always fell between $64.04\%$ and $64.75\%$ (Figure S5). We also ran 9 sequential CIFAR10 trials using a shorter training period to demonstrate reproducibility of the early training curve (Figure S6).

As a control study, we tested how sensitive both Sparse Combo Net and SVD Combo Net were to the stabilization condition on the interconnection matrix ($\mathbf{L}$ in Theorem 6). To do this we constrained the hidden-to-hidden weights of the networks to satisfy their respective contraction conditions. However, instead of constraining $\mathbf{L}$ in a way that ensures contraction of the overall system, we set $\mathbf{L} = \mathbf{B}$ and did not constrain $\mathbf{B}$ during training. In both cases we found that this hurt network performance on the permuted seqMNIST task, demonstrating the utility of the contraction condition. For the $24 \times 32$ Sparse Combo Network we saw a decrease from $96.7\%$ test accuracy to $47.0\%$ test accuracy. For the $24 \times 32$ SVD Combo Network we saw a decrease from $94.9\%$ to $94.56\%$. The disparity in performance decrease makes sense when considering that the hidden-to-hidden weights of SVD Combo Network are trainable, while those of the Sparse Combo Network are not. Whatever instabilities are introduced by the lack of constraint on the inter-subnetwork connections cannot be adequately compensated for in the Sparse Combo Network.

## 4    DISCUSSION

Most work on stability of task-trained RNNs has focused on *single* RNNs. Here we leverage tools from nonlinear control theory to derive novel single-RNN stability conditions which enable the recursive construction of stable *assemblies* of RNNs. In particular we show that certain stability conditions for individual RNNs allow for a simple parameterization of connections between these RNNs that automatically preserves stability during training. We then show that these modular 'network of networks' repeatedly perform better than existing stable RNNs on key sequence classification tasks such as permuted seqMNIST and seqCIFAR10. We also provide control studies that show the stabilizing parameterization of connections between subnetwork RNNs is important for high performance.

There are numerous future directions enabled by this work. For example, Theorem 7 suggests that a less restrictive contraction condition on $\mathbf{W}$ in terms of the eigenvalues of the symmetric part is possible and desirable. Furthermore, the beneficial impact of sparsity on training these stable models suggests a potential avenue for additional experimental work – in particular adding a regularizing sparsity term during training. As 'network of network' approaches are becoming increasingly popular, our methodology is relevant to a variety of task types, including reinforcement learning applications. Moreover, because sparsity and modularity have important theoretical and empirical implications in neuroscience (Kozachkov et al., 2020; Slotine & Liu, 2012) we expect to be able to apply the present work to those topics as well.

Ultimately, our work represents a step forward in understanding the stability properties of recurrent neural networks. Stability is a fundamental property of dynamical systems, and is inextricably linked to concepts such as generalization, control, predictability, and robustness. Therefore, as systems trained with deep learning become more modular, complex, and integrated into our lives, understanding the conditions under which these systems are stable will become increasingly important. Furthermore, it will be necessary to identify training techniques that can reliably generate provably stable RNNs with minimal performance loss. We provide a proof-of-concept of the use of combination networks to achieve this goal, and show that sparsity enables additional performance gains.

**Acknowledgements** This work benefited from stimulating discussions with Michael Happ and Quang-Cuong Pham.

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

APPENDIX

## A1 CONTRACTION MATH

It can be shown that the non-autonomous system

$$\dot{\mathbf{x}} = \mathbf{f}(\mathbf{x}, t)$$

is contracting if there exists a metric $\mathbf{M}(\mathbf{x}, t) = \mathbf{\Theta}(\mathbf{x}, t)^T \mathbf{\Theta}(\mathbf{x}, t) \succ 0$ such that uniformly

$$\dot{\mathbf{M}} + \mathbf{M}\mathbf{J} + \mathbf{J}^T \mathbf{M} \preceq -\beta \mathbf{M}$$

where $\mathbf{J} = \frac{\partial \mathbf{f}}{\partial \mathbf{x}}$ and $\beta > 0$. For more details see the main reference (Lohmiller & Slotine, 1998). Similarly, a non-autonomous discrete-time system

$$\mathbf{x}_{t+1} = \mathbf{f}(\mathbf{x}_t, t)$$

is contracting if

$$\mathbf{J}^T \mathbf{M}_{t+1} \mathbf{J} - \mathbf{M}_t \preceq -\beta \mathbf{M}_t$$

### A1.1 FEEDBACK AND HIERARCHICAL COMBINATIONS

Consider two systems, independently contracting in constant metrics $\mathbf{M}_1$ and $\mathbf{M}_2$, which are combined in feedback:

$$\begin{aligned} \dot{\mathbf{x}} &= \mathbf{f}(\mathbf{x}, t) + \mathbf{B}\mathbf{y} \\ \dot{\mathbf{y}} &= \mathbf{g}(\mathbf{y}, t) + \mathbf{G}\mathbf{x} \end{aligned} \qquad \text{(Feedback Combination)}$$

If the following relationship between $\mathbf{B}, \mathbf{G}, \mathbf{M}_1$, and $\mathbf{M}_2$ is satisfied:

$$\mathbf{B} = -\mathbf{M}_1^{-1} \mathbf{G}^T \mathbf{M}_2$$

then the combined system is contracting as well. This may be seen as a special case of the feedback combination derived in (Tabareau & Slotine, 2006). The situation is even simpler for hierarchical combinations. Consider again two systems, independently contracting in some metrics, which are combined in hierarchy:

$$\begin{aligned} \dot{\mathbf{x}} &= \mathbf{f}(\mathbf{x}, t) \\ \dot{\mathbf{y}} &= \mathbf{g}(\mathbf{y}, t) + \mathbf{h}(\mathbf{x}, t) \end{aligned} \qquad \text{(Hierarchical Combination)}$$

where $\mathbf{h}(\mathbf{x}, t)$ is a function with *bounded* Jacobian. Then this combined system is contracting in a diagonal metric, as shown in (Lohmiller & Slotine, 1998). By recursion, this extends to hierarchies of arbitrary depth.

## A2 TWO DIFFERENT RNNS

Note that in neuroscience, the variable $\mathbf{x}$ in equation (1) is typically thought of as a vector of neural membrane potentials. It was shown in (Miller & Fumarola, 2012) that the RNN (1) is equivalent via an affine transformation to another commonly used RNN model,

$$\tau \dot{\mathbf{y}} = -\mathbf{y} + \phi(\mathbf{W}\mathbf{y} + \mathbf{b}(t)) \qquad (4)$$

where the variable $\mathbf{y}$ is interpreted as a vector of firing rates, rather than membrane potentials. The two models are related by the transformation $\mathbf{x} = \mathbf{W}\mathbf{y} + \mathbf{b}$, which yields

$$\tau \dot{\mathbf{x}} = \mathbf{W}(-\mathbf{y} + \phi(\mathbf{W}\mathbf{y} + \mathbf{b})) + \tau \dot{\mathbf{b}} = -\mathbf{x} + \mathbf{W}\phi(\mathbf{x}) + \mathbf{v}$$

where $\mathbf{v} \equiv \mathbf{b} + \tau\dot{\mathbf{b}}$. Thus $\mathbf{b}$ is a low-pass filtered version of $\mathbf{v}$ (or conversely, $\mathbf{v}$ may be viewed as a first order prediction of $\mathbf{b}$) and the contraction properties of the system are unaffected by the affine transformation. Note that the above equivalence holds even in the case where $\mathbf{W}$ is not invertible. In this case, the two models are proven to be equivalent, provided that $\mathbf{b}(0)$ and $\mathbf{y}(0)$ satisfy certain conditions–which are always possible to satisfy (Miller & Fumarola, 2012). Therefore, any contraction condition derived for the $x$ (or $y$) system automatically implies contraction of the other system. We exploit this freedom freely throughout the paper.

## A3  SPARSE COMBO NET SUPPLEMENTARY MATERIALS

All networks described in the main text were trained using a single GPU on Google Colab. An exported Colab notebook with the code to replicate all experiments is provided in the supplementary attachment.

To obtain the metric necessary for training the linear connections, scipy.integrate.quad was used with default settings to solve for $\mathbf{M}$ in the equation $-\mathbf{I} = \mathbf{MW} + \mathbf{W}^T\mathbf{M}$, as described in the main text. This is done by integrating $e^{\mathbf{W}^T t}\mathbf{Q}e^{\mathbf{W}t}dt$ from 0 to $\infty$. For efficiency reasons, and due to the guaranteed existence of a diagonal metric in the case of Theorem 1, integration was only performed to solve for the diagonal elements of $\mathbf{M}$. Therefore a check was added prior to training to confirm that the initialized network indeed satisfied Theorem 1 with metric $\mathbf{M}$. However, it was never triggered by our initialization method.

Initial training hyperparameter tuning was done primarily with $10 \times 16$ combination networks on the permuted seqMNIST task, starting with settings based on existing literature on this task, and verifying promising settings using a $15 \times 16$ network. Initialization settings were held the same throughout, matching what was later done for the size comparison trials (described below). The results of all of the attempted trials are reported in Table S1. Once hyperparameters were decided upon, the trials reported on in the main text began.

To report on the number of trainable parameters, we used the following formula:

$\frac{n^2 - M*C^2}{2} + i*n + n*o + n + o$

Where $n$ is the total number of units in the $M \times C$ combination network, $o$ is the total number of output nodes for the task, and $i$ is the total number of input nodes for the task. Thus for the $16 \times 32$ networks highlighted here, we have 129034 trainable parameters for the MNIST tasks, and 130058 trainable parameters for sequential CIFAR10.

Note that the naive estimate for the number of trainable parameters would be $n^2 + i*n + n*o + n + o$, corresponding to the number of weights in $\mathbf{L}$, the number of weights in the feedforward linear input layer, the number of weights in the feedforward linear output layer, and the bias terms for the input and output layers, respectively. However, because of the combination property constraints on $\mathbf{L}$, only the lower triangular portion of a block off-diagonal matrix is actually trained, and $\mathbf{L}$ is then defined in terms of this matrix and the metric $\mathbf{M}$. Thus we subtract $M * C^2$ to remove the block diagonal portions corresponding to nonlinear RNN components, and then divide by 2 to obtain only the lower half.

After training was completed, we inspected the state of all networks described in the main text, pulling both the nonlinear ($\mathbf{W}$) and linear ($\mathbf{L}$) weight matrices from both initialization time and the final model. For $\mathbf{W}$, we confirmed it did not change over training, and inspected the max real part of the eigenvalues of $|\mathbf{W}|$ in accordance with Theorem 1. The densest tested matrices tended to have $\lambda_{max}(|\mathbf{W}|) > 0.9$, while the sparsest ones tended to have $\lambda_{max}(|\mathbf{W}|) < 0.1$. For $\mathbf{L}$, we checked the maximum element and the maximum singular value before and after training. In general, both went up over the course of training, but by a modest amount.

### A3.1  NETWORK SIZE AND MODULARITY COMPARISON

For the size comparison trials (Figure 5A), the nonlinear RNN weights were set by drawing uniformly from between $-0.4$ and $0.4$ with $40\%$ density using scipy.sparse.random, and then zeroing out the diagonal entries. These settings were chosen because they resulted in $\sim 1\%$ of 16 by 16

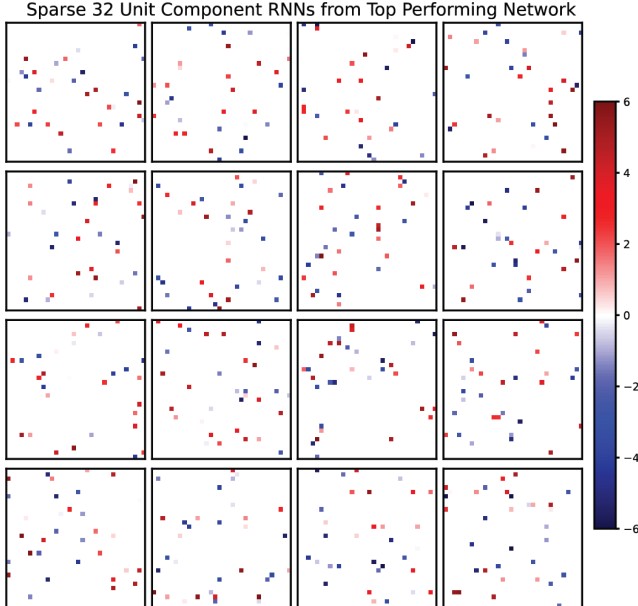

Figure S1: Weight matrices for each of the 32 unit non-linear component RNNs that were used in the best performing $16 \times 32$ network on permuted sequential MNIST.

weight matrices meeting the Theorem 1 condition. During initialization only the matrices meeting this condition were kept, finishing when the desired number of component RNNs had been set - producing a block diagonal $\mathbf{W}$ like pictured in Figure 3. This same initialization process was used throughout our experiments.

First, we held static the number of units and initialization settings for each component RNN, and tested the effect of changing the number of components in the combination network. 1, 3, 5, 10, 15, 20, 22, 25, and 30 component RNNs were tested in this experiment (Table S2). Increasing the number of components initially lead to great improvements in test accuracy, but had diminishing returns - test accuracy consistently hit $\sim 93\%$ with a large enough number of modules, but neither loss nor accuracy showed meaningful improvement past the $22 \times 16$ network (Figure S2A). Interestingly, early training loss and accuracy became substantially worse once the number of components increased past a certain point, falling from 70% to 43% epoch 1 test accuracy between the $22 \times 16$ and $30 \times 16$ networks.

To better understand how the modularity of the combination networks affects performance, the next experiment held the number of total units constant at 352, selected due to the prior success of the $22 \times 16$ network, and tested different allocations of these units amongst component RNNs. Thus $1 \times 352$, $11 \times 32$, $44 \times 8$, and $88 \times 4$ networks were trained to compare against the $22 \times 16$ (Table S3). Increasing the modularity improved performance to a point, with the $44 \times 8$ network resulting in final test accuracy of $94.44\%$, while conversely the $11 \times 32$ resulted in decreased test accuracy (Figure S2B). However, the $88 \times 4$ network was unable to learn, and a $352 \times 1$ network would theoretically just be a scaled linear anti-symmetric network.

Because larger networks require different sparsity settings to meet the Theorem 1 condition, these were not held constant between trials in the modularity comparison experiment (Figure 5B), but rather selected in the same way between trials - looking for settings that keep density and scalar balanced and result in $\sim 1\%$ of the matrices meeting the condition. The scalar was applied after sampling non-zero entries from a uniform distribution between -1 and 1. The resulting settings were 7.5% density and 0.077 scalar for 352 unit component RNN, 26.5% density and 0.27 scalar for 32 unit component RNN, 60% density and 0.7 scalar for 8 unit component RNN, and 100% density and 1.0 scalar for 4 unit component RNN.

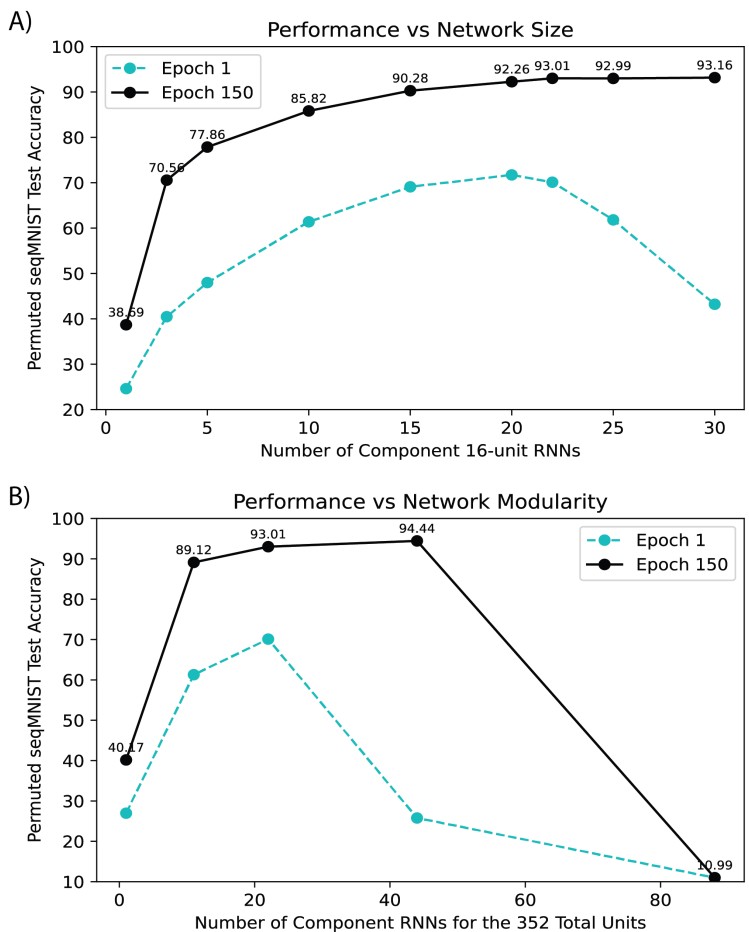

Figure S2: Performance of Sparse Combo Nets on the Permuted seqMNIST task by combination network size. We test the effects on final and first epoch test accuracy of both total network size and network modularity. The former is assessed by varying the number of subnetworks while each subnetwork is fixed at 16 units (A), and the latter by varying the distribution of units across different numbers of subnetworks with the total sum of units in the network fixed at 352 (B). Note that these experiments were run prior to optimizing the sparsity initialization settings. Experiments on total network size were later repeated with the final sparsity settings (Figure 5). The results of both the size experiments are consistent.

### A3.2 SPARSITY SETTINGS COMPARISON

Density and scalar settings for the component nonlinear RNNs were initially chosen for each network size using the percentage of random networks that met the Theorem 1 condition. For scalar $s$, a component network would have non-zero entries sampled uniformly between $-s$ and $s$.

When we began experimenting with sparsity in the initialization, we split the previously described scalar setting into two different scalars - one applied before a random matrix was checked against the Theorem 1 condition, and one applied after a matrix was selected. Of course the latter must be $\leq 1$ to guarantee stability is preserved. The scalar was separated out after we noticed that at $5\%$ density, random 32 by 32 weight matrices met the condition roughly $1\%$ of the time whether the scalar was 10 or 100000 - $\sim 85\%$ of sampled matrices using scalar 10 would continue to meet the condition even if multiplied by a factor of 10000. Therefore we wanted a mechanism that could bias selection towards matrices that are stable due to their sparsity and not due to magnitude constraints, while still keeping the elements to a reasonable size for training purposes.

Ultimately, both sparsity and magnitude had a clear effect on performance (Figure S3). Increases in both had a positive correlation with accuracy and loss through most parameters tested, seemingly driven by the increase in non-zero element magnitude enabled by sparsity. Best test accuracy overall was 96.79%, which was obtained by both a $16 \times 32$ network with $5\%$ density and entries between -5 and 5, and a $16 \times 32$ network with $3.3\%$ density and entries between -6 and 6. The latter also achieved the best epoch 1 test accuracy observed of $86.79\%$.

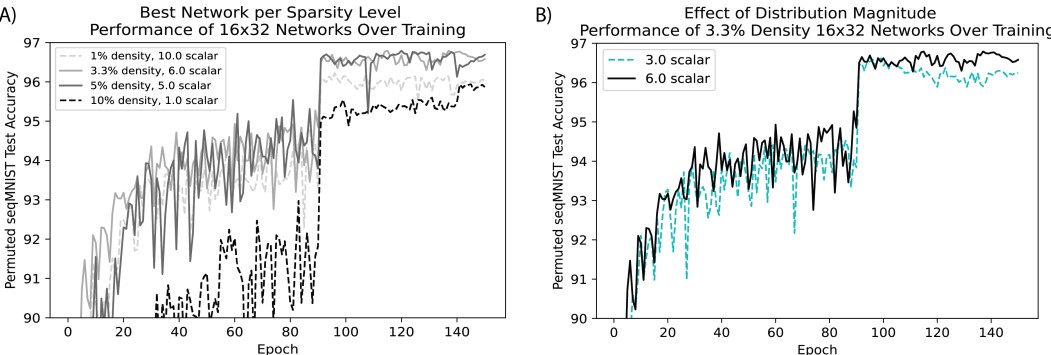

Figure S3: Permuted seqMNIST performance by component RNN initialization settings. Test accuracy is plotted over the course of training for four $16 \times 32$ networks with different density levels and entry magnitudes (A), highlighting the role of sparsity in network performance. Test accuracy is then plotted over the course of training for two 3.3% density $16 \times 32$ networks with different entry magnitudes (B), to demonstrate the role of the scalar. When the magnitude becomes too high however, performance is out of view of the current axis limits.

### A3.3 REPEATABILITY

To further improve performance once network settings were explored on permuted seqMNIST, an extended training run was tested on the best performing option. Settings were kept the same as above using a $3.3\%$ density $16 \times 32$ network, except training now ran for over 200 epochs, with just a single learning rate cut occurring after epoch 200 (exact number of epochs varied based on runtime limit). This experiment was repeated four times and resulted in $96.94\%$ best test accuracy (Figure S4).

We performed a similar repetition task for the seqCIFAR10 task. To avoid hitting runtime limits, we performed three trials where each network was trained for 200 epochs with learning rate cuts at epochs 140 and 190. All networks exceeded $64\%$ test accuracy, with the best performing network achieving $64.75\%$ (Figure S5). We also trained a larger number of networks over a smaller number of epochs (Figure S6), as well as explored additional hyperparameter tuning on this task (Table S6).

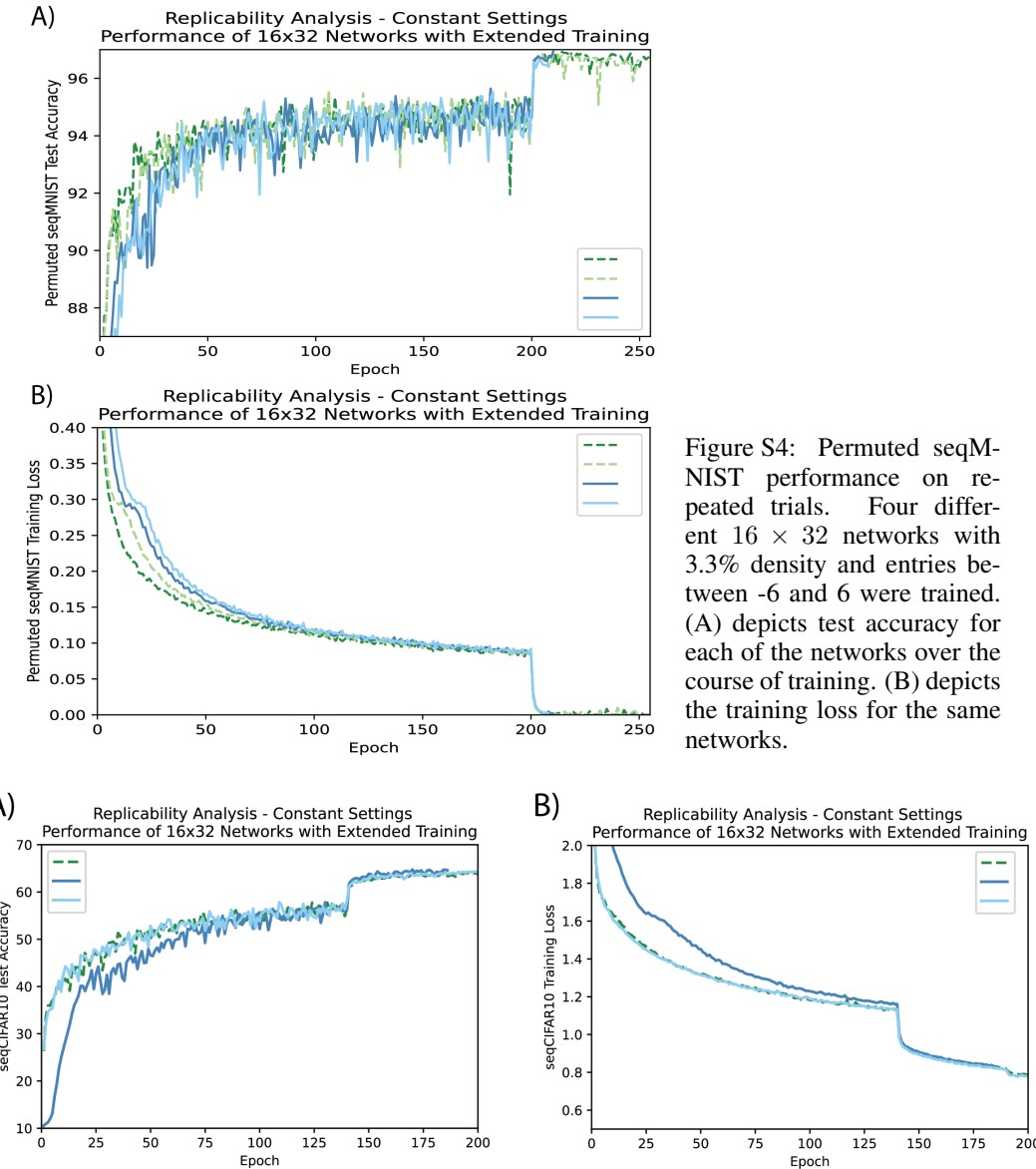

Figure S4: Permuted seqM-NIST performance on repeated trials. Four different $16 \times 32$ networks with 3.3% density and entries between -6 and 6 were trained. (A) depicts test accuracy for each of the networks over the course of training. (B) depicts the training loss for the same networks.

Figure S5: seqCIFAR10 performance on repeated trials. Three different $16 \times 32$ networks with 3.3% density and entries between -6 and 6 were trained for 200 epochs, with learning rate divided by 10 after epochs 140 and 190. (A) depicts test accuracy for each of the networks over the course of training. (B) depicts the training loss for the same networks.

### A3.4 TABLES OF RESULTS BY TRIAL

Table S1 shows all trials run on permuted sequential MNIST before beginning the more systematic experiments reported on in the main text, presented in chronological order. Notably, our networks did not require an extensive hyperparameter tuning process.

Tables S2 and S3 report additional details on the size and modularity experiments (Figure S2).

Tables S4 and S5 report additional details on the sparsity experiments described in the main text (Figure 4), including results of all relevant trials, as some were left out of the main text for brevity.

Table S6 reports the results of all trials of different hyperparameters on the sequential CIFAR10 task, in chronological order.

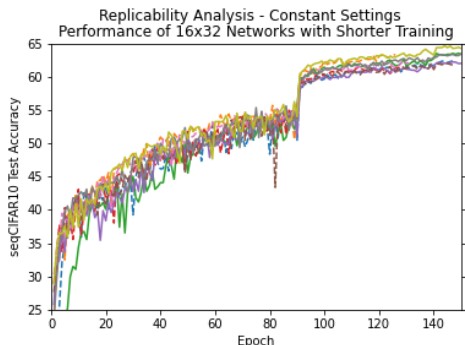

Figure S6: seqCIFAR10 performance on repeated trials with shorter training (done to complete more trials). Nine different $16 \times 32$ networks with 3.3% density and entries between -6 and 6 were set up to train for 150 epochs, with learning rate divided by 10 after epochs 90 and 140. Most of these networks hit runtime limit before completing, however they all got through at least 100 epochs and all had test accuracy exceed 61%. This figure depicts test accuracy for each of the networks over the course of training.

Finally, Table S7 reports the results of our pilot trial on introducing sparsity into the linear feedback connection matrix.

| Size | Epochs | Adam WD | Initial LR | LR Schedule | Final Test Acc. |
|------|--------|---------|-----------|-------------|-----------------|
| $10 \times 16$ | 150 | 5e-5 | 5e-3 | 0.1 after 91 | 84% |
| $10 \times 16$ | 150 | 1e-5 | 1e-2 | 0.1 after 50,100 | 85% |
| $15 \times 16$ | 150 | 2e-4 | 5e-3 | 0.1 after 50,100 | 84% |
| $10 \times 16$ | 150 | 2e-4 | 1e-2 | 0.5 every 10 | 81% |
| $10 \times 16$ | 200 | 2e-4 | 1e-2 | 0.5 after 10 then every 30 | 81% |
| $10 \times 16$ | 171* | 5e-5 | 1e-2 | 0.75 after 10,20,60,100 then every 15 | 84% |
| $15 \times 16$ | 179* | 1e-5 | 1e-3 | 0.1 after 100,150 | 90% |

Table S1: Training hyperparameter tuning trials, presented in chronological order. * indicates that training was cut short by the 24 hour Colab runtime limit. LR Schedule describes the scalar the learning rate was multiplied by, and at what epochs. The best performing network is highlighted, and represents the training settings we used throughout most of the main text.

| Size | Final Test Acc. | Epoch 1 Test Acc. | Final Train Loss |
|---|---|---|---|
| $1 \times 16$ | 38.69% | 24.61% | 1.7005 |
| $3 \times 16$ | 70.56% | 40.47% | 0.9033 |
| $5 \times 16$ | 77.86% | 47.99% | 0.7104 |
| $10 \times 16$ | 85.82% | 61.38% | 0.4736 |
| $15 \times 16$ | 90.28% | 69.09% | 0.3156 |
| $20 \times 16$ | 92.26% | 71.72% | 0.2392 |
| $22 \times 16$ | 93.01% | 70.11% | 0.2073 |
| $25 \times 16$ | 92.99% | 61.81% | 0.2017 |
| $30 \times 16$ | 93.16% | 43.21% | 0.1991 |

Table S2: Results for combination networks containing different numbers of component 16-unit RNNs. Training hyperparameters and network initialization settings were kept the same across all trials.

| Size | Final Test Acc. | Epoch 1 Test Acc. | Final Train Loss |
|---|---|---|---|
| $1 \times 352$ | 40.17% | 26.97% | 1.662 |
| $11 \times 32$ | 89.12% | 61.29% | 0.3781 |
| $22 \times 16$ | 93.01% | 70.11% | 0.2073 |
| $44 \times 8$ | 94.44% | 25.78% | 0.1500 |
| $88 \times 4$ | 10.99% | 10.99% | 2E+35 |

Table S3: Results for different distributions of 352 total units across a combination network. This number was chosen based on prior $22 \times 16$ network performance. For each component RNN size tested, the same procedure was used to select appropriate density and scalar settings.

| Size | Density | Scalar | Final Test Acc. | Epoch 1 Test Acc. | Final Train Loss |
|---|---|---|---|---|---|
| $11 \times 32$ | 26.5% | 0.27 | 89.12% | 61.29% | 0.3781 |
| $11 \times 32$ | 10% | 1.0 | 94.86% | 70.67% | 0.1278 |
| $22 \times 16$ | 40% | 0.4 | 93.01% | 70.11% | 0.2073 |
| $22 \times 16$ | 20% | 1.0 | 95.27% | 76.58% | 0.0924 |
| $22 \times 16$ | 10% | 1.0 | 94.26% | 71.53% | 0.1425 |
| $44 \times 8$ | 60% | 0.7 | 94.44% | 25.78% | 0.1500 |
| $44 \times 8$ | 50% | 1.0 | 95.05% | 30.52% | 0.1180 |

Table S4: Results for different initialization settings - varying sparsity and magnitude of the component RNNs for different network sizes.

| Density | Pre-select Scalar | Post-select Scalar | Final Test Acc. | Epoch 1 Test Acc. | Final Train Loss |
|---|---|---|---|---|---|
| 10% | 1.0 | 1.0 | 95.87% | 73.67% | 0.074 |
| 5% | 10.0 | 0.1 | 95.11% | 73.10% | 0.1311 |
| 5% | 10.0 | 0.2 | 96.15% | 82.50% | 0.0051 |
| 5% | 10.0 | 0.5 | 96.69% | 75.76% | 0.0001 |
| 5% | 6.0 | 1.0 | 96.41% | 21.55% | 3.3E-5 |
| 5% | 7.5 | 1.0 | 16.75% | 11.39% | 3068967 |
| 3.3% | 30.0 | 0.1 | 96.24% | 83.89% | 0.0005 |
| 3.3% | 30.0 | 0.2 | 96.54% | 86.79% | 4E-5 |
| 1% | 10.0 | 1.0 | 96.04% | 81.2% | 0.0001 |

Table S5: Further optimizing the sparsity settings for high performance using a $16 \times 32$ network. The final scalar is the product of the pre-selection and post-selection scalars. Note that the 5% density and 7.5 scalar network was killed after 18 epochs due to exploding gradient.

| Density | Pre-select Scalar | Post-select Scalar | Epochs | Adam WD | Initial LR | LR Schedule | Best Test Acc. |
|---|---|---|---|---|---|---|---|
| 3.3% | 30 | 0.2 | 150 | 1e-5 | 1e-3 | 0.1 after 90,140 | 64.63% |
| 3.3% | 30 | 0.2 | 34* | 1e-5 | 5e-3 | 0.1 after 90,140 | 35.42% |
| 5% | 6 | 1 | 150 | 1e-5 | 1e-3 | 0.1 after 90,140 | 60.9% |
| 5% | 10 | 0.5 | 150 | 1e-5 | 1e-4 | 0.1 after 90,140 | 54.86% |
| 3.3% | 30 | 0.2 | 150 | 1e-5 | 5e-4 | 0.1 after 90,140 | 61.83% |
| 3.3% | 30 | 0.2 | 200 | 1e-6 | 2e-3 | 0.1 after 140,190 | 62.31% |
| 3.3% | 30 | 0.2 | 186* | 1e-5 | 1e-3 | 0.1 after 140,190 | 64.75% |
| 3.3% | 30 | 0.2 | 132* | 1e-6 | 1e-3 | 0.1 after 140,190 | 62.31% |
| 5% | 10 | 0.5 | 195* | 1e-5 | 1e-3 | 0.1 after 140,190 | 64.68% |

Table S6: Additional hyperparameter tuning for the CIFAR10 task, presented in chronological order. * indicates that training was cut short by the 24 hour Colab runtime limit, or in the case of high learning rate was killed intentionally due to exploding gradient. LR Schedule describes the scalar the learning rate was multiplied by, and at what epochs. The best performing network is highlighted.

| Size | Feedback Density | Epochs | Best Overall Test Acc. | Best Test Acc. Through 85 Epochs |
|---|---|---|---|---|
| 24×32 | 100% | 86 | 52.7% | 52.7% |
| 24×32 | 75% | 88 | 56.49% | 56.48% |
| 24×32 | 66.6% | 89 | 58.84% | 58.84% |
| 24×32 | 50% | 124 | 65.14% | 58.01% |
| 24×32 | 33.3% | 129 | 61.86% | 56.05% |
| 24×32 | 25% | 92 | 54.26% | 50.54% |
| 24×32 | 0% | 130 | 39.8% | 38.38% |
| 16×32 | 100% | 150 | 64.63% | 55.82% |

Table S7: Results from pilot testing on the sparsity of negative feedback connections in a $24 \times 32$ Sparse Combo Net. Feedback Density refers to the percentage of possible subnetwork pairings that were trained in negative feedback, while the remaining inter-network connections were held at 0. All networks were trained with the same 150 epoch training paradigm as mentioned in the main text, but were stopped after hitting a 24 hour runtime limit. Decreasing Feedback Density is a promising path towards further improving performance as the size of Sparse Combo Nets is scaled.

## A4 PROOFS FOR MAIN RESULTS

### A4.1 PROOF OF THEOREM 1

Our first theorem is motivated by the observation that if the y-system is to be interpreted as a vector of firing rates, it must stay positive for all time. For a linear, time-invariant system with positive states, diagonal stability is equivalent to stability. Therefore a natural question is if diagonal stability of a linearized y-system implies anything about stability of the nonlinear system. More formally, given an excitatory neural network (i.e $\forall ij, W_{ij} \geq 0$), if the linear system

$$\dot{\mathbf{x}} = -\mathbf{x} + g\mathbf{W}\mathbf{x}$$

is stable, then there exists a positive diagonal matrix P such that:

$$\mathbf{P}(g\mathbf{W} - \mathbf{I}) + (g\mathbf{W} - \mathbf{I})^T\mathbf{P} \prec 0$$

The following theorem shows that the nonlinear system (1) is indeed contracting in metric $\mathbf{P}$, and extends this result to a more general $\mathbf{W}$ by considering only the magnitudes of the weights.

**Theorem 1.** *Let $|\mathbf{W}|$ denote the matrix formed by taking the element-wise absolute value of $\mathbf{W}$. If there exists a positive, diagonal $\mathbf{P}$ such that:*

$$\mathbf{P}(g|\mathbf{W}| - \mathbf{I}) + (g|\mathbf{W}| - \mathbf{I})^T\mathbf{P} \prec 0$$

*then equation 1 is contracting in metric $\mathbf{P}$. Moreover, if $W_{ii} \leq 0$, then $|W|_{ii}$ may be set to zero to reduce conservatism.*

This condition is particularly straightforward in the common special case where the network does not have any self weights, with the leak term driving stability. While it can be applied to a more general $\mathbf{W}$, the condition will of course not be met if the network was relying on highly negative values on the diagonal of $\mathbf{W}$ for linear stability. As demonstrated by counterexample in the proof of Theorem 1, it can be impossible to use the same metric $\mathbf{P}$ for the nonlinear RNN in such cases.

Theorem 1 allows many weight matrices with low magnitudes or a generally sparse structure to be verified as contracting in the nonlinear system equation 1, by simply checking a linear stability condition (as linear stability is equivalent to diagonal stability for Metzler matrices too (Narendra & Shorten, 2010)).

Beyond verifying contraction, Theorem 1 actually provides a metric, with little need for additional computation. Not only is it of inherent interest that the same metric can be shared across systems in this case, it is also of use in machine learning applications, where stability certificates are becoming increasingly necessary. Critically, it is feasible to enforce the condition during training via L2 regularization on $\mathbf{W}$. More generally, there are a variety of systems of interest that meet this condition but do not meet the well-known maximum singular value condition, including those with a hierarchical structure.

*Proof.* Consider the differential, quadratic Lyapunov function:

$$V = \delta\mathbf{x}^T\mathbf{P}\delta\mathbf{x}$$

where $\mathbf{P} \succ 0$ is diagonal. The time derivative of $V$ is:

$$\dot{V} = 2\delta\mathbf{x}^T\mathbf{P}\dot{\delta\mathbf{x}} = 2\delta\mathbf{x}^T\mathbf{P}\mathbf{J}\delta\mathbf{x} = -2\delta\mathbf{x}^T\mathbf{P}\delta\mathbf{x} + 2\delta\mathbf{x}^T\mathbf{P}\mathbf{W}\mathbf{D}\delta\mathbf{x}$$

where $\mathbf{D}$ is a diagonal matrix such that $\mathbf{D}_{ii} = \frac{d\phi_i}{dx} \geq 0$. We can upper bound the quadratic form on the right as follows:

$$\delta\mathbf{x}^T\mathbf{P}\mathbf{W}\mathbf{D}\delta\mathbf{x} = \sum_{ij} P_iW_{ij}D_j\delta x_i\delta x_j \leq$$

$$\sum_i P_iW_{ii}D_i|\delta x_i|^2 + \sum_{ij,i\neq j} P_i|W_{ij}|D_j|\delta x_i||\delta x_j| \leq g|\delta\mathbf{x}|^T\mathbf{P}|\mathbf{W}||\delta\mathbf{x}|$$

If $W_{ii} \leq 0$, the term $P_iW_{ii}D_i|\delta x_i|^2$ contributes non-positively to the overall sum, and can therefore be set to zero without disrupting the inequality. Now using the fact that $\mathbf{P}$ is positive and diagonal, and therefore $\delta\mathbf{x}^T\mathbf{P}\delta\mathbf{x} = |\delta\mathbf{x}|^T\mathbf{P}|\delta\mathbf{x}|$, we can upper bound $\dot{V}$ as:

$$\dot{V} \leq |\delta\mathbf{x}|^T(-2\mathbf{P} + \mathbf{P}|\mathbf{W}| + |\mathbf{W}|\mathbf{P})|\delta\mathbf{x}| = |\delta\mathbf{x}|^T[(\mathbf{P}(|\mathbf{W}| - \mathbf{I}) + (|\mathbf{W}|^T - \mathbf{I})\mathbf{P})]|\delta\mathbf{x}|$$

where $|W|_{ij} = |W_{ij}|$, and $|W|_{ii} = 0$ if $W_{ii} \leq 0$ and $|W|_{ii} = |W_{ii}|$ if $W_{ii} > 0$. This completes the proof. Note that $\mathbf{W} - \mathbf{I}$ is Metzler, and therefore will be Hurwitz stable if and only if $\mathbf{P}$ exists (Narendra & Shorten, 2010).

It is also worth noting that highly negative diagonal values in $\mathbf{W}$ will prevent the same metric $\mathbf{P}$ from being used for the nonlinear system. Therefore the method used in this proof cannot feasibly be adapted to further relax the treatment of the diagonal part of $\mathbf{W}$. The intuitive reason behind this is that in the symmetric part of the Jacobian, $\frac{\mathbf{PWD} + \mathbf{DW}^T\mathbf{P}}{2} - \mathbf{P}$, the diagonal self weights will also be scaled down by small $\mathbf{D}$, while the leak portion $-\mathbf{P}$ remains untouched by $\mathbf{D}$. Now we actually demonstrate a counterexample, presenting a $2 \times 2$ symmetric Metzler matrix $\mathbf{W}$ that is contracting in the identity in the linear system, but cannot be contracting *in the identity* in the nonlinear system equation 1:

$$\mathbf{W} = \begin{bmatrix} -9 & 2.5 \\ 2.5 & 0 \end{bmatrix}$$

To see that it is not possible for the more general nonlinear system with these weights to be contracting in the identity, take $\mathbf{D} = \begin{bmatrix} 0 & 0 \\ 0 & 1 \end{bmatrix}$. Now

$$(\mathbf{WD})_{sym} - \mathbf{I} = \begin{bmatrix} -1 & 1.25 \\ 1.25 & -1 \end{bmatrix}$$

which has a positive eigenvalue of $\frac{1}{4}$.

$\square$

## A4.2   Proof of Theorem 2

While regularization may push networks towards satisfying Theorem 1, strictly enforcing the condition during optimization is not straightforward. This motivates the rest of our theorems, which derive contraction results for specially structured weight matrices. Unlike Theorem 1, these results have direct parameterizations which can easily be plugged into modern optimization libraries.

**Theorem 2.** *If $\mathbf{W} = \mathbf{W}^T$ and $g\mathbf{W} \prec \mathbf{I}$, then (1) is contracting.*

When $\mathbf{W}$ is symmetric, (1) may be seen as a continuous-time Hopfield network. Continuous-time Hopfield networks with symmetric weights were recently shown to be closely related to Transformer architectures (Krotov & Hopfield, 2020; Ramsauer et al., 2020). Specifically, the dot-product attention rule may be seen as a discretization of the continuous-time Hopfield network with softmax activation function (Krotov & Hopfield, 2020). Our results here provide a simple sufficient (and nearly necessary, see above remark) condition for global exponential stability of a given *trajectory* for the Hopfield network. In the case where the input into the network is constant, this trajectory is a fixed point. Moreover, each trajectory associated with a unique input is guaranteed to be unique. Finally, we note that our results are flexible with respect to activation functions so long as they satisfy the slope-restriction condition. This flexibility may be useful when, for example, considering recent work showing that standard activation functions may be advantageously replaced by attention mechanisms (Dai et al., 2020).

*Proof.* We begin by writing $\mathbf{W} = \mathbf{R} - \mathbf{P}$ for some unknown $\mathbf{R} = \mathbf{R}^T$ and $\mathbf{P} = \mathbf{P}^T \succ 0$. The approach of this proof is to show by construction that the condition $g\mathbf{W} \prec \mathbf{I}$ implies the existence of an $\mathbf{R}$ and $\mathbf{P}$ such that the system is contracting in metric $\mathbf{P}$. We consider the $y$ version of the RNN, which as discussed above is equivalent to the $x$ version via an affine transformation.

Consider the contraction condition:

$$-2\mathbf{M} + \mathbf{MDW} + \mathbf{W}^T\mathbf{DM} \preceq -\beta\mathbf{M}$$

with $\beta > 0$. Substituting in the definitions of $\mathbf{W}$ and $\mathbf{M}$, this condition becomes:

$$-2\mathbf{P} + \mathbf{PD}(\mathbf{R} - \mathbf{P}) + (\mathbf{R} - \mathbf{P})\mathbf{DP} \preceq -\beta\mathbf{P}$$

Simplifying the terms and collecting them all on one side, the above may be written as:

$$(\beta - 2)\mathbf{P} + \mathbf{RDP} + \mathbf{PDR} - 2\mathbf{PDP} \preceq 0$$

via the Schur complement, the above term will be satisfied if:

$$(2 - \beta)\mathbf{P} - \mathbf{RDP}(2\mathbf{PDP})^{-1}\mathbf{PDR} =$$

$$(2 - \beta)\mathbf{P} - \frac{1}{2}(\mathbf{RDR}) \succeq (2 - \beta)\mathbf{P} - \frac{g}{2}(\mathbf{RR}) \succeq 0$$

We continue by setting $\mathbf{P} = \gamma^2\mathbf{RR}$ with $\gamma^2 = \frac{g}{2(2-\beta)}$, so that the above inequality is satisfied. At this point, we have shown that if $\mathbf{W}$ can be written as:

$$\mathbf{W} = \mathbf{R} - \gamma^2\mathbf{RR}$$

then (1) is contracting in metric $\mathbf{M} = \gamma^2\mathbf{RR}$. What remains to be shown is that if the condition:

$$g\mathbf{W} - \mathbf{I} \prec 0$$

Is satisfied, then this implies the existence of an $\mathbf{R}$ such that the above is true. To show that this is indeed the case, assume that:

$$\frac{1}{4\gamma^2}\mathbf{I} - \mathbf{W} \succeq 0$$

Substituting in the definition of $\gamma$, this is just the statement that:

$$\frac{2(2 - \beta)}{4g}\mathbf{I} - \mathbf{W} \succeq 0$$

Setting $\beta = 2\lambda > 0$, this yields:

$$(1 - \lambda)\mathbf{I} \succeq g\mathbf{W}$$

Since $\mathbf{W}$ is orthogonal, we have the eigendecomposition:

$$\frac{1}{4\gamma^2}\mathbf{I} - \mathbf{W} = \mathbf{V}(\frac{1}{4\gamma^2}\mathbf{I} - \mathbf{\Lambda})\mathbf{V}^T$$

where $\mathbf{V}^T\mathbf{V} = \mathbf{I}$ and $\mathbf{\Lambda}$ is a diagonal matrix containing the eigenvalues of $\mathbf{W}$. Denote the symmetric square-root of this expression as $\mathbf{S}$:

$$\mathbf{S} = \mathbf{V}\sqrt{(\frac{1}{4\gamma^2}\mathbf{I} - \mathbf{\Lambda})}\mathbf{V}^T = \mathbf{S}^T$$

Which implies that:

$$\frac{1}{4\gamma^2}\mathbf{I} - \mathbf{W} = \mathbf{S}^T\mathbf{S}$$

We now define $\mathbf{R}$ in terms of $\mathbf{S}$ as follows:

$$\mathbf{R} = \frac{1}{\gamma}\mathbf{S} + \frac{1}{2\gamma^2}\mathbf{I}$$

Which means that:

$$\frac{1}{4\gamma^2}\mathbf{I} - \mathbf{W} = (\gamma\mathbf{R} - \frac{1}{2\gamma}\mathbf{I})(\gamma\mathbf{R} - \frac{1}{2\gamma}\mathbf{I})$$

Expanding out the right side, we get:

$$\frac{1}{4\gamma^2}\mathbf{I} - \mathbf{W} = \gamma^2\mathbf{RR} - \mathbf{R} + \frac{1}{4\gamma^2}\mathbf{I}$$

Subtracting $\frac{1}{4\gamma^2}\mathbf{I}$ from both sides yields:

$$\mathbf{W} = \mathbf{R} - \gamma^2\mathbf{RR}$$

As desired.

$\square$

## A4.3    PROOF OF THEOREM 3

**Theorem 3.** *If there exists positive diagonal matrices* $\mathbf{P}_1$ *and* $\mathbf{P}_2$*, as well as* $\mathbf{Q} = \mathbf{Q}^T \succ 0$ *such that*

$$\mathbf{W} = -\mathbf{P}_1\mathbf{Q}\mathbf{P}_2$$

*then (1) is contracting in metric* $\mathbf{M} = (\mathbf{P}_1\mathbf{Q}\mathbf{P}_1)^{-1}$*.*

*Proof.* Consider again a differential Lyapunov function:

$$V = \delta\mathbf{x}^T\mathbf{M}\delta\mathbf{x}$$

the time derivative is equal to:

$$\dot{V} = -2V + \delta\mathbf{x}^T\mathbf{M}\mathbf{W}\mathbf{D}\delta\mathbf{x}$$

Substituting in the definitions of $\mathbf{W}$ and $\mathbf{M}$, we get:

$$\dot{V} = -2V - \delta\mathbf{x}^T\mathbf{P}_1^{-1}\mathbf{P}_2\mathbf{D}\delta\mathbf{x} \leq -2V$$

Therefore $V$ converges exponentially to zero.

$\square$

## A4.4    PROOF OF THEOREM 4

**Theorem 4.** *If* $g\mathbf{W} - \mathbf{I}$ *is triangular and Hurwitz, then (1) is contracting in a diagonal metric.*

Note that in the case of a triangular weight matrix, the system (1) may be seen as a feedforward (i.e hierarchical) network. Therefore, this result follows from the combination properties of contracting systems. However, our proof provides a means of explicitly finding a metric for this system.

*Proof.* Without loss of generality, assume that $\mathbf{W}$ is lower triangular. This implies that $W_{ij} = 0$ if $i \leq j$. Now consider the generalized Jacobian:

$$\mathbf{F} = -\mathbf{I} + \boldsymbol{\Gamma}\mathbf{W}\mathbf{D}\boldsymbol{\Gamma}^{-1}$$

with $\boldsymbol{\Gamma}$ diagonal and $\Gamma_i = \epsilon^i$ where $\epsilon > 0$. Because $\boldsymbol{\Gamma}$ is diagonal, the generalized Jacobian is equal to:

$$\mathbf{F} = -\mathbf{I} + \boldsymbol{\Gamma}\mathbf{W}\boldsymbol{\Gamma}^{-1}\mathbf{D}$$

Now note that:

$$(\boldsymbol{\Gamma}\mathbf{W}\boldsymbol{\Gamma}^{-1})_{ij} = \epsilon^i W_{ij}\epsilon^{-j} = W_{ij}\epsilon^{i-j}$$

Where $i \leq j$, we have $W_{ij} = 0$ by assumption. Therefore, the only nonzero entries are where $i \geq j$. This means that by making $\epsilon$ arbitrarily small, we can make $\boldsymbol{\Gamma}\mathbf{W}\boldsymbol{\Gamma}^{-1}$ approach a diagonal matrix with $W_{ii}$ along the diagonal. Therefore, if:

$$\max_i gW_{ii} - 1 < 0$$

the nonlinear system is contracting. Since $\mathbf{W}$ is triangular, $W_{ii}$ are the eigenvalues of $\mathbf{W}$, meaning that this condition is equivalent to $g\mathbf{W} - \mathbf{I}$ being Hurwitz.

$\square$

## A4.5    PROOF OF THEOREM 5

**Theorem 5.** *If there exists a positive diagonal matrix* $\mathbf{P}$ *such that:*

$$g^2\mathbf{W}^T\mathbf{P}\mathbf{W} - \mathbf{P} \prec 0$$

*then (1) is contracting in metric* $\mathbf{P}$*.*

Note that this is equivalent to the discrete-time diagonal stability condition developed in (Revay & Manchester, 2020), for a constant metric. Note also that when $\mathbf{M} = \mathbf{I}$, Theorem 5 is identical to checking the maximum singular value of $\mathbf{W}$, a previously established condition for stability of equation 1. However a much larger set of weight matrices are found via the condition when $\mathbf{M} = \mathbf{P}$ instead.

*Proof.* Consider the generalized Jacobian:

$$\mathbf{F} = \mathbf{P}^{1/2}\mathbf{J}\mathbf{P}^{-1/2} = -\mathbf{I} + \mathbf{P}^{1/2}\mathbf{W}\mathbf{P}^{-1/2}\mathbf{D}$$

where $\mathbf{D}$ is a diagonal matrix with $\mathbf{D}_{ii} = \frac{d\phi_i}{dx_i}$. Using the subadditivity of the matrix measure $\mu_2$ of the generalized Jacobian we get:

$$\mu_2(\mathbf{F}) \leq -1 + \mu_2(\mathbf{P}^{1/2}\mathbf{W}\mathbf{P}^{-1/2}\mathbf{D})$$

Now using the fact that $\mu_2(\cdot) \leq ||\cdot||_2$ we have:

$$\mu_2(\mathbf{F}) \leq -1 + ||\mathbf{P}^{1/2}\mathbf{W}\mathbf{P}^{-1/2}\mathbf{D})||_2 \leq -1 + g||\mathbf{P}^{1/2}\mathbf{W}\mathbf{P}^{-1/2}||_2$$

Using the definition of the 2-norm, imposing the condition $\mu_2(\mathbf{F}) \leq 0$ may be written:

$$g^2\mathbf{W}^T\mathbf{P}\mathbf{W} - \mathbf{P} \prec 0$$

which completes the proof.

$\square$

### A4.6 Proof of Theorem 6

**Theorem 6** (Network of Networks). *Consider a collection of $p$ subnetwork RNNs governed by equation 1. Assume that these RNNs each have hidden-to-hidden weight matrices $\{\mathbf{W}_1, \ldots, \mathbf{W}_p\}$ and are independently contracting in metrics $\{\mathbf{M}_1, \ldots, \mathbf{M}_p\}$. Define the block matrices $\tilde{\mathbf{W}} \equiv BlockDiag(\mathbf{W}_1, \ldots, \mathbf{W}_p)$ and $\tilde{\mathbf{M}} \equiv BlockDiag(\mathbf{M}_1, \ldots, \mathbf{M}_p)$, as well as the overall state vector $\tilde{\mathbf{x}}^T \equiv (\mathbf{x}_1^T \cdots \mathbf{x}_2^T)$. Then the following 'network of networks' is globally contracting in metric $\tilde{\mathbf{M}}$:*

$$\tau\dot{\tilde{\mathbf{x}}} = -\tilde{\mathbf{x}} + \tilde{\mathbf{W}}\phi(\tilde{\mathbf{x}}) + \mathbf{u}(t) + \mathbf{L}\tilde{\mathbf{x}}$$
$$\mathbf{L} \equiv \mathbf{B} - \tilde{\mathbf{M}}^{-1}\mathbf{B}^T\tilde{\mathbf{M}} \tag{5}$$

*Where $\mathbf{B}$ is an arbitrary square matrix.*

*Proof.* To see that this overall system is contracting in constant metric $\tilde{\mathbf{M}}$, consider the corresponding *differential Lyapunov equation* (Lohmiller & Slotine, 1998):

$$\lambda_{max}(\beta\tilde{\mathbf{M}} + \tilde{\mathbf{M}}\mathbf{J} + \mathbf{J}^T\tilde{\mathbf{M}}) \leq$$
$$0 \tag{6}$$
$$\lambda_{max}(2(\beta\tilde{\mathbf{M}} - \tilde{\mathbf{M}} + \tilde{\mathbf{M}}\tilde{\mathbf{W}}\tilde{\mathbf{D}})_s) + \lambda_{max}(\tilde{\mathbf{M}}\mathbf{B} - \mathbf{B}^T\tilde{\mathbf{M}}) \leq 0$$

where $\mathbf{J}$ is the Jacobian of equation 5 and $D_{ii} = \phi'(\tilde{x}_i)$ is a diagonal matrix containing the slopes of the activation functions. The last inequality follows from the original assumption that the individual subnetworks are contracting in metrics $\{\mathbf{M}_1, \ldots, \mathbf{M}_p\}$ with rate $\beta$. $\square$

### A4.7 Proof of Theorem 7

**Theorem 7.** *Let $\mathbf{D}$ be a positive, diagonal matrix with $D_{ii} = \frac{d\phi_i}{dx_i}$, and let $\mathbf{P}$ be an arbitrary, positive diagonal matrix. If:*

$$(g\mathbf{W} - \mathbf{I})\mathbf{P} + \mathbf{P}(g\mathbf{W}^T - \mathbf{I}) \preceq -c\mathbf{P}$$

*and*

$$\dot{\mathbf{D}} - cg^{-1}\mathbf{D} \preceq -\beta\mathbf{D}$$

*for $c, \beta > 0$, then (1) is contracting in metric $\mathbf{D}$ with rate $\beta$.*

*Proof.* Consider the differential, quadratic Lyapunov function:

$$V = \delta\mathbf{x}^T\mathbf{PD}\delta\mathbf{x}$$

where $\mathbf{D} \succ 0$ is as defined above. The time derivative of $V$ is:

$$\dot{V} = \delta\mathbf{x}^T\mathbf{P}\dot{\mathbf{D}}\delta\mathbf{x} + \delta\mathbf{x}^T(-2\mathbf{PD} + \mathbf{PDWD} + \mathbf{DW}^T\mathbf{DP})\delta\mathbf{x}$$

The second term on the right can be factored as:

$$\begin{aligned}
\delta\mathbf{x}^T(-2\mathbf{PD} + \mathbf{PDWD} + \mathbf{DW}^T\mathbf{DP})\delta\mathbf{x} = \\
\delta\mathbf{x}^T\mathbf{D}(-2\mathbf{PD}^{-1} + \mathbf{PW} + \mathbf{W}^T\mathbf{P})\mathbf{D}\delta\mathbf{x} \leq \\
\delta\mathbf{x}^T\mathbf{D}(-2\mathbf{P}g^{-1} + \mathbf{PW} + \mathbf{W}^T\mathbf{P})\mathbf{D}\delta\mathbf{x} = \\
\delta\mathbf{x}^T\mathbf{D}[\mathbf{P}(\mathbf{W} - g^{-1}\mathbf{I}) + (\mathbf{W}^T - g^{-1}\mathbf{I})\mathbf{P}]\mathbf{D}\delta\mathbf{x} \leq \\
-cg^{-1}\delta\mathbf{x}^T\mathbf{PD}^2\delta\mathbf{x}
\end{aligned}$$

where the last inequality was obtained by substituting in the first assumption above. Combining this with the expression for $\dot{V}$, we have:

$$\dot{V} \leq \delta\mathbf{x}^T\mathbf{P}\dot{\mathbf{D}}\delta\mathbf{x} - cg^{-1}\delta\mathbf{x}^T\mathbf{PD}^2\delta\mathbf{x}$$

Substituting in the second assumption, we have:

$$\dot{V} \leq \delta\mathbf{x}^T\mathbf{P}(\dot{\mathbf{D}} - cg^{-1}\mathbf{D}^2)\delta\mathbf{x} \leq -\beta\delta\mathbf{x}^T\mathbf{PD}\delta\mathbf{x} = -\beta V$$

and thus $V$ converges exponentially to 0 with rate $\beta$. $\qquad\square$

### A4.8 Proof of Theorem 8

**Theorem 8.** *Satisfaction of the condition*

$$g\mathbf{W}_{sym} - \mathbf{I} \prec 0$$

*is **NOT** sufficient to show global contraction of the general nonlinear RNN (1) in any constant metric. High levels of antisymmetry in $\mathbf{W}$ can make it impossible to find such a metric, which we demonstrate via a $2 \times 2$ counterexample of the form*

$$\mathbf{W} = \begin{bmatrix} 0 & -c \\ c & 0 \end{bmatrix}$$

*with $c \geq 2$.*

Note that $g\mathbf{W}_{sym} - \mathbf{I} = g\frac{\mathbf{W}+\mathbf{W}^T}{2} - \mathbf{I} \prec 0$ is equivalent to the condition for contraction of the system with *linear* activation in the identity metric.

The main intuition behind this counterexample is that high levels of antisymmetry can prevent a constant metric from being found in the nonlinear system. This is because $\mathbf{D}$ is a diagonal matrix with values between 0 and 1, so the primary functionality it can have in the symmetric part of the

Jacobian is to downweight the outputs of certain neurons selectively. In the extreme case of all 0 or 1 values, we can think of this as selecting a subnetwork of the original network, and taking each of the remaining neurons to be single unit systems receiving input from the subnetwork. For a given static configuration of $\mathbf{D}$ (think linear gains), this is a hierarchical system that will be stable if the subnetwork is stable. But as $\mathbf{D}$ can evolve over time when a nonlinearity is introduced, we would need to find a constant metric that can serve completely distinct hierarchical structures simultaneously - which is not always possible.

Put in terms of matrix algebra, D can zero out columns of $\mathbf{W}$, but not their corresponding rows. So for a given weight pair $w_{ij}, w_{ji}$, which has entry in $\mathbf{W}_{sym} = \frac{w_{ij}+w_{ji}}{2}$, if $D_i = 0$ and $D_j = 1$, the $i, j$ entry in $(\mathbf{WD})_{sym}$ will be guaranteed to have lower magnitude if the signs of $w_{ij}$ and $w_{ji}$ are the same, but guaranteed to have higher magnitude if the signs are different. Thus if the linear system would be stable based on magnitudes alone $\mathbf{D}$ poses no real threat, but if the linear system requires antisymmetry to be stable, $\mathbf{D}$ can make proving contraction quite complicated (if possible at all).

*Proof.* The nonlinear system is globally contracting in a *constant* metric if there exists a symmetric, positive definite $\mathbf{M}$ such that the symmetric part of the Jacobian for the system, $(\mathbf{MWD})_{sym} - \mathbf{M}$ is negative definite uniformly. Therefore $(\mathbf{MWD})_{sym} - \mathbf{M} \prec 0$ must hold for all possible $\mathbf{D}$ if $\mathbf{M}$ is a constant metric the system *globally* contracts in with any allowed activation function, as some combination of settings to obtain a particular $\mathbf{D}$ can always be found.

Thus to prove the main claim, we present here a simple 2-neuron system that is contracting in the identity metric with linear activation function, but can be shown to have no $\mathbf{M}$ that simultaneously satisfies the $(\mathbf{MWD})_{sym} - \mathbf{M} \prec 0$ condition for two different possible $\mathbf{D}$ matrices.

To begin, take

$$\mathbf{W} = \begin{bmatrix} 0 & -2 \\ 2 & 0 \end{bmatrix}$$

Note that any off-diagonal magnitude $\geq 2$ would work, as this is the point at which $\frac{1}{2}$ of one of the weights (found in $\mathbf{W}_{sym}$ when the other is zeroed) will have magnitude too large for $(\mathbf{WD})_{sym} - \mathbf{I}$ to be stable.

Looking at the linear system, we can see it is contracting in the identity because

$$\mathbf{W}_{sym} - \mathbf{I} = \begin{bmatrix} -1 & 0 \\ 0 & -1 \end{bmatrix} \prec 0$$

Now consider $(\mathbf{MWD})_{sym} - \mathbf{M}$ with $\mathbf{D}$ taking two possible values of

$$\mathbf{D}_1 = \begin{bmatrix} 1 & 0 \\ 0 & 0 \end{bmatrix} \quad and \quad \mathbf{D}_2 = \begin{bmatrix} 0 & 0 \\ 0 & 1 \end{bmatrix}$$

We want to find some symmetric, positive definite $\mathbf{M} = \begin{bmatrix} a & m \\ m & b \end{bmatrix}$ such that $(\mathbf{MWD}_1)_{sym} - \mathbf{M}$ and $(\mathbf{MWD}_2)_{sym} - \mathbf{M}$ are both negative definite.

Working out the matrix multiplication, we get

$$(\mathbf{MWD}_1)_{sym} - \mathbf{M} = \begin{bmatrix} 2m - a & b - m \\ b - m & -b \end{bmatrix}$$

and

$$(\mathbf{MWD}_2)_{sym} - \mathbf{M} = \begin{bmatrix} -a & -(a+m) \\ -(a+m) & -2m - b \end{bmatrix}$$

We can now check necessary conditions for negative definiteness on these two matrices, as well as for positive definiteness on $\mathbf{M}$, to try to find an $\mathbf{M}$ that will satisfy all these conditions simultaneously. In this process we will reach a contradiction, showing that no such $\mathbf{M}$ can exist.

A necessary condition for positive definiteness in a real, symmetric $n \times n$ matrix $\mathbf{X}$ is $x_{ii} > 0$, and for negative definiteness $x_{ii} < 0$. Another well known necessary condition for definiteness of a real symmetric matrix is $|x_{ii} + x_{jj}| > |x_{ij} + x_{ji}| = 2|x_{ij}| \;\; \forall i \neq j$. See (Weisstein) for more info on these conditions.

Thus we will require $a$ and $b$ to be positive, and can identify the following conditions as necessary for our 3 matrices to all meet the requisite definiteness conditions:

$$2m < a \tag{7}$$

$$-2m < b \tag{8}$$

$$|2m - (a + b)| > 2|b - m| \tag{9}$$

$$|-2m - (a + b)| > 2|a + m| \tag{10}$$

Note that the necessary condition for $\mathbf{M}$ to be PD, $a + b > 2|m|$, is not listed, as it is automatically satisfied if equation 7 and equation 8 are.

It is easy to see that if $m = 0$, conditions equation 9 and equation 10 will result in the contradictory conditions $a > b$ and $b > a$ respectively, so we will require a metric with off-diagonal elements. To make the absolute values easier to deal with, we will check $m > 0$ and $m < 0$ cases independently.

First we take $m > 0$. By condition equation 7 we must have $a > 2m$, so between that and knowing the signs of all unknowns are positive, we can reduce many of the absolute values. Condition equation 9 becomes $a + b - 2m > |2b - 2m|$, and condition equation 10 becomes $a + b + 2m > 2a + 2m$, which is equivalent to $b > a$. If $b > a$ we must also have $b > m$, so condition equation 9 further reduces to $a + b - 2m > 2b - 2m$, which is equivalent to $a > b$. Therefore we have again reached contradictory conditions.

A very similar approach can be applied when $m < 0$. Using condition equation 8 and the known signs we reduce condition equation 9 to $2|m| + a + b > 2b + 2|m|$, i.e. $a > b$. Meanwhile condition equation 10 works out to $a + b - 2|m| > 2a - 2|m|$, i.e. $b > a$.

Therefore it is impossible for a single constant $\mathbf{M}$ to accommodate both $\mathbf{D}_1$ and $\mathbf{D}_2$, so that no constant metric can exist for $\mathbf{W}$ to be contracting in when a nonlinearity is introduced that can possibly have derivative reaching both of these configurations. One real world example of such a nonlinearity is ReLU. Given a sufficiently high negative input to one of the units and a sufficiently high positive input to the other, $\mathbf{D}$ can reach one of these configurations. The targeted inputs could then flip at any time to reach the other configuration.

An additional condition we could impose on the activation function is to require it to be a strictly increasing function, so that the activation function derivative can never actually reach 0. We will now show that a very similar counterexample applies in this case, by taking

$$\mathbf{D}_{1*} = \begin{bmatrix} 1 & 0 \\ 0 & \epsilon \end{bmatrix} \quad and \quad \mathbf{D}_{2*} = \begin{bmatrix} \epsilon & 0 \\ 0 & 1 \end{bmatrix}$$

Note here that the $\mathbf{W}$ used above produced a $(\mathbf{WD})_{sym} - \mathbf{I}$ that just barely avoided being negative definite with the original $\mathbf{D}_1$ and $\mathbf{D}_2$, so we will have to increase the values on the off-diagonals a

bit for this next example. In fact anything with magnitude larger than 2 will have some $\epsilon > 0$ that will cause a constant metric to be impossible, but for simplicity we will now take

$$\mathbf{W}_* = \begin{bmatrix} 0 & -4 \\ 4 & 0 \end{bmatrix}$$

Note that with $\mathbf{W}_*$, even just halving one of the off-diagonals while keeping the other intact will produce a $(\mathbf{WD})_{sym} - \mathbf{I}$ that is not negative definite. Anything less than halving however will keep the identity metric valid. Therefore, we expect that taking $\epsilon$ in $\mathbf{D}_{1*}$ and $\mathbf{D}_{2*}$ to be in the range $0.5 \geq \epsilon > 0$ will also cause issues when trying to obtain a constant metric.

We will now actually show via a similar proof to the above that $\mathbf{M}$ is impossible to find for $\mathbf{W}_*$ when $\epsilon \leq 0.5$. This result is compelling because it not only shows that $\epsilon$ does not need to be a particularly small value, but it also drives home the point about antisymmetry - the larger in magnitude the antisymmetric weights are, the larger the $\epsilon$ where we will begin to encounter problems.

Working out the matrix multiplication again, we now get

$$(\mathbf{MW}_*\mathbf{D}_{1*})_{sym} - \mathbf{M} = \begin{bmatrix} 4m - a & 2b - m - 2a\epsilon \\ b - m - 2a\epsilon & -4m\epsilon - b \end{bmatrix}$$

and

$$(\mathbf{MW}_*\mathbf{D}_{2*})_{sym} - \mathbf{M} = \begin{bmatrix} 4m\epsilon - a & -(2a + m - 2b\epsilon) \\ -(2a + m - 2b\epsilon) & -4m - b \end{bmatrix}$$

Resulting in two new main necessary conditions:

$$|4m - a - b - 4m\epsilon| > 2|2b - m - 2a\epsilon| \tag{11}$$

$$|4m\epsilon - a - b - 4m| > 2|2a + m - 2b\epsilon| \tag{12}$$

As well as new conditions on the diagonal elements:

$$4m - a < 0 \tag{13}$$

$$-4m - b < 0 \tag{14}$$

We will now proceed with trying to find $a, b, m$ that can simultaneously meet all conditions, setting $\epsilon = 0.5$ for simplicity.

Looking at $m = 0$, we can see again that $\mathbf{M}$ will require off-diagonal elements, as condition equation 11 is now equivalent to the condition $a + b > |4b - 2a|$ and condition equation 12 is similarly now equivalent to $a + b > |4a - 2b|$.

Evaluating these conditions in more detail, if we assume $4b > 2a$ and $4a > 2b$, we can remove the absolute value and the conditions work out to the contradicting $3a > 3b$ and $3b > 3a$ respectively. As an aside, if $\epsilon > 0.5$, this would no longer be the case, whereas with $\epsilon < 0.5$, the conditions would be pushed even further in opposite directions.

If we instead assume $2a > 4b$, this means $4a > 2b$, so the latter condition would still lead to $b > a$, contradicting the original assumption of $2a > 4b$. $2b > 4a$ causes a contradiction analogously. Trying $4b = 2a$ will lead to the other condition becoming $b > 2a$, once again a contradiction. Thus a diagonal $\mathbf{M}$ is impossible

So now we again break down the conditions into $m > 0$ and $m < 0$ cases, first looking at $m > 0$. Using condition equation 13 and knowing all unknowns have positive sign, condition equation 11 reduces to $a + b - 2m > |4b - 2(a + m)|$ and condition equation 12 reduces to $a + b + 2m > |4a - 2(b - m)|$. This looks remarkably similar to the $m = 0$ case, except now condition equation 11 has $-2m$ added to both sides (inside the absolute value), and condition equation 12 has $2m$ added to both sides in the same manner. If $4b > 2(a + m)$ the $-2m$ term on each side will simply cancel,

and similarly if $4a > 2(b - m)$ the $+2m$ terms will cancel, leaving us with the same contradictory conditions as before.

Therefore we check $2(a + m) > 4b$. This rearranges to $2a > 2(2b - m) > 2(b - m)$, so that from condition equation 12 we get $b > a$. Subbing condition equation 13 in to $2(a + m) > 4b$ gives $8b < 4a + 4m < 5a$ i.e. $b < \frac{5}{8}a$, a contradiction. The analogous issue arises if trying $2(b - m) > 4a$. Trying $2(a + m) = 4b$ gives $m = 2b - a$, which in condition equation 12 results in $5b - a > |6a - 6b|$, while in condition equation 13 leads to $5a > 8b$, so equation 12 can further reduce to $5b - a > 6a - 6b$ i.e. $11b > 7a$. But $b > \frac{7}{11}a$ and $b < \frac{5}{8}a$ is a contradiction. Thus there is no way for $m > 0$ to work.

Finally, trying $m < 0$, we now use condition equation 14 and the signs of the unknowns to reduce condition equation 11 to $a+b+2|m| > |4b-2(a-|m|)|$ and condition equation 12 to $a+b-2|m| > |4a - 2(b+|m|)|$. These two conditions are clearly directly analogous to in the $m > 0$ case, where $b$ now acts as $a$ with condition equation 14 being $b > 4|m|$. Therefore the proof is complete.

$\square$

## A5   CODE FOR MODELS USED IN EXPERIMENTS

```
# class definitions for the sparse combo networks

class rnnAssemblyCell_Thm1(LightningModule):
    '''
    Pytorch module for training the following system:
        tau*dx/dt = -x + W*phi(x) + L*x+ u(t)
    where tau > 0, phi is a nonlinearity, W is block diagonal, L is some 'contracting' combination matrix and u is some input.

    '''

    def __init__(self, input_size, hidden_sizes, output_size, alpha, A, density, pre_select_mult, post_select_mult):
        super(rnnAssemblyCell_Thm1, self).__init__()
        self.input_size = input_size
        self.hidden_sizes = hidden_sizes
        self.hidden_size = int(np.sum(hidden_sizes))
        self.output_size = output_size
        self.alpha = alpha

        # initialize linear input and output layers, to be trained, along with biases
        self.weight_ih = nn.Parameter(torch.normal(0,1/np.sqrt(self.hidden_size),(self.hidden_size, self.input_size)))
        self.weight_ho = nn.Parameter(torch.normal(0,1/np.sqrt(self.hidden_size),(self.output_size, self.hidden_size)))
        self.bias_oh = nn.Parameter(torch.normal(0,1/np.sqrt(self.hidden_size),(1,self.output_size)))
        self.bias_hh = nn.Parameter(torch.normal(0,1/np.sqrt(self.hidden_size),(1,self.hidden_size)))

        #self.register_buffer("V_mask", create_mask_given_A(torch.eye(len(ns)),ns).bool())
        # specify W and M here based on the random initialization mentioned
        W, M = generate_initial_W_M(self.hidden_sizes, density, pre_select_mult, post_select_mult)
        M_inv = torch.inverse(M)

        self.register_buffer("W",W)
        self.register_buffer("M", M)
        self.register_buffer("M_inv", M_inv)

        # L contains the connections between subsystems. this will be trained.
        L_mask = create_mask_given_A(A,self.hidden_sizes).bool()
        self.register_buffer("L_mask", L_mask)

        self.L_train = nn.Parameter(self.L_mask*torch.normal(0,1/np.sqrt(np.mean(self.hidden_sizes)),(np.sum(self.hidden_sizes), np.sum(self.hidden_sizes)))

    def forward(self, input):

        L_masked = self.L_train*self.L_mask

        state = torch.zeros((input.shape[0],self.hidden_size),device = self.device)
        state = state.type_as(state)

        inputs = input.unbind(1)

        #outputs = []

        for i in range(len(inputs)):

            fx = -state + F.relu(state @ self.W.T + inputs[i] @ (self.weight_ih.T) + self.bias_hh) + state @ (L_masked.T - (self.M @ L_masked) @ self.M_inv

            state =  state + self.alpha*fx

            hy = state @ (self.weight_ho.T)

            #outputs += [hy]

        return hy, state
```

Figure S7: Pytorch Lightning code for Sparse Combo Net Cell

```python
#define network
class rnnAssembly_SV(LightningModule):
    '''
    Pytorch module for training the following system:
        tau*dx/dt = -x + W*phi(x) + L*x+ u(t)
    where tau > 0, phi is a nonlinearity, W is block diagonal, L is some 'contracting' combination matrix and u is some input.

    '''

    def __init__(self, input_size, hidden_sizes, output_size,alpha,A):
        super(rnnAssembly_SV, self).__init__()
        self.input_size = input_size
        self.hidden_sizes = hidden_sizes
        self.hidden_size = int(np.sum(hidden_sizes))
        ns = hidden_sizes
        self.output_size = output_size
        self.alpha = alpha

        #input to hidden weights
        self.weight_ih = nn.Parameter(torch.normal(0,1/np.sqrt(self.hidden_size),(self.hidden_size, self.input_size)))
        #hidden to output weights
        self.weight_ho = nn.Parameter(torch.normal(0,1/np.sqrt(self.hidden_size),(self.output_size, self.hidden_size)))

        #mask to modularize overall network
        self.register_buffer("V_mask", create_mask_given_A(torch.eye(len(ns)),ns).bool())

        #parameterization of hidden-to-hidden networks
        self.U = nn.Parameter(torch.eye(self.hidden_size))
        self.V = nn.Parameter(torch.eye(self.hidden_size))
        self.Sigma = nn.Parameter(torch.ones((self.hidden_size,)))
        self.Phi = nn.Parameter(torch.normal(1,1/np.sqrt(self.hidden_size),(self.hidden_size,)))
        self.sing_val_eps = 1e-3
        self.register_buffer("L_mask", create_mask_given_A(A,ns).bool())
        self.register_buffer("S_offset",(self.sing_val_eps)*torch.ones(self.Sigma.shape[0]))

        #trainable inter-subnetwork weights
        self.L_train = nn.Parameter(self.L_mask*torch.normal(0,1/np.sqrt(np.mean(ns)),(np.sum(ns), np.sum(ns))))

        #biases
        self.bias_oh = nn.Parameter(torch.normal(0,1/np.sqrt(self.hidden_size),(1,self.output_size)))
        self.bias_hh = nn.Parameter(torch.normal(0,1/np.sqrt(self.hidden_size),(1,self.hidden_size)))

    def forward(self, input):

        #mask the left and right orth 'stem' matrices to produce diagonal-block structure
        U_masked = self.U*self.V_mask
        V_masked = self.V*self.V_mask

        #take skew-symmetric part of these matrices
        #exponentiate skew-symmetric part to produce left and right orthogonal matrices
        O1 = torch.matrix_exp(U_masked - U_masked.T)
        O2 = torch.matrix_exp(V_masked - V_masked.T)

        #get diagonal of singular value matrix, with elements between [0,1)
        S = torch.exp(-self.Sigma**2 -self.S_offset)

        #construct the weight matrix to be contracting in metric determined by Phi
        W = torch.diag_embed(self.Phi) @ (O1 @ torch.diag_embed(S) @ O2.T) @ torch.diag_embed(1/self.Phi)

        #get metric and metric inverse
        M = torch.diag_embed(1/(self.Phi**2))
        M_inv = torch.diag_embed(self.Phi**2)

        #get mask for network-to-network coupling
        L_masked = self.L_train*self.L_mask

        #initialize hidden state of the network
        inputs = input.unbind(1)
        state = torch.zeros((input.shape[0],self.hidden_size),device = self.device)
        state = state.type_as(state)

        #propagate input through the dynamics and store outputs
        for i in range(len(inputs)):
            fx = -state + F.relu(state @ W.T + inputs[i] @ (self.weight_ih.T) + self.bias_hh) + state @ (L_masked.T - (M @ L_masked) @ M_inv)
            state = state + self.alpha*fx
        hy = state @ (self.weight_ho.T)
        return hy, state
```

Figure S8: Pytorch Lightning code for SVD Combo Net cell.

## A5.1 NETWORK DIAGRAMS

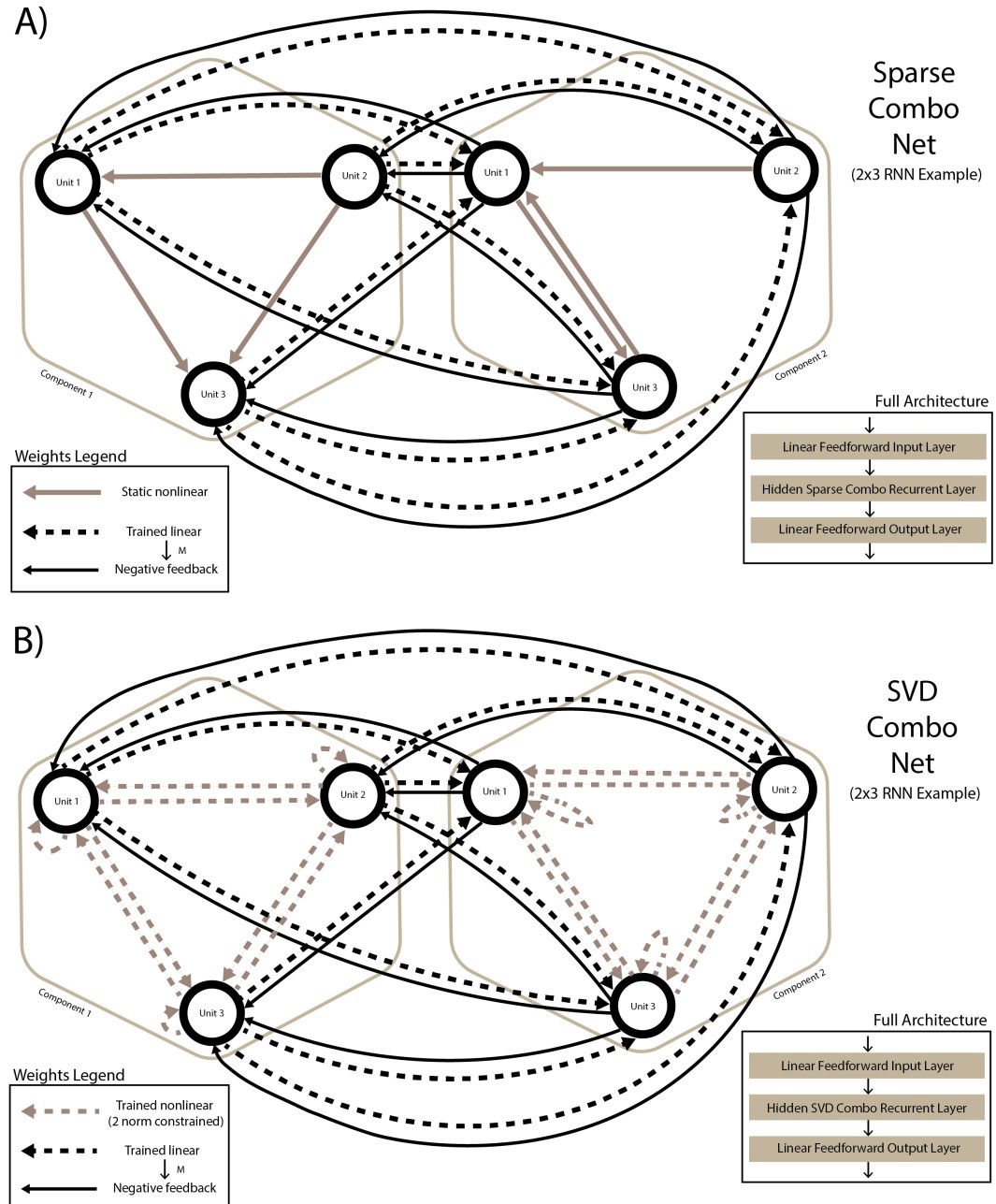

Figure S9: Detailed architecture diagrams for Sparse Combo Net (A) and SVD Combo Net (B).

