# OpenReview forum: "Recursive Construction of Stable Assemblies of Recurrent Neural Networks"
_ICLR.cc/2022/Conference — ICLR 2022 Submitted_

### Official Review · Reviewer_RFwT · 2021-10-26

**Correctness:** 2
**Technical Novelty And Significance:** 3
**Empirical Novelty And Significance:** 2
**Recommendation:** 6
**Confidence:** 2

**Main Review:**

## Theoretical results

The theorems 1-5 constitute an evolutionary step in the understanding the conditions of stability. The authors also show a counterexample for the common belief that linear contraction leads a sufficient condition for non-linear stability. These results are then (partly) used to construct provably stable RNN combinations.

## Experimental claims

The experimental section, while interesting, seems to lack a main takeaway. Also some very dubious choice of reporting in the table.

* The authors claim that for the result shown on the Tab. 1, they run the Perm-MNIST trial 4 times and the results fall between 96.65 and 96.94. I was frankly shocked to find that they choose to only report 96.94 on Tab.1! Running the same experiment multiple times and only reporting the best case scenario is not good practice and leads to misunderstanding at best. I would recommend that the authors report mean + variance. (If one were to only report the best case, one could get much better performance than is achievable on average by running the experiment many many times.)

* Some claims are backed up by only a single data point. For example, the claim that increased modularity benefits performance to some point is only backed up by the fact that 44x8 performs better than 22x16 in Sec. 3.2.1. To draw a significant conclusion and demonstrate a trend, the authors can perhaps look at 50x7 and 39x9.

* In general, more data points and error bars would help convince the reader that the conclusions are real and not flukes.

* The results of 'performance vs network size' and 'performance vs modularity' for Sec. 3.1 and 3.2 are opposite each other. In 3.1 increased size makes the network better monotonically but in 3.2 it is inverted U shape. Similarly in 3.2 modularity makes the performance better monotonically, but it is inverted U in 3.1. What is the conclusion to be drawn here?

* In general, the experiment section is a little hard to read and can use a summary of the main conclusions and clearly demarcated paragraphs and sections of the experiment that demonstrate each point.



### Clarity and other minor points

* The paper is at times very clear and at times very confusing. For example, the discussion of stability and contraction are clear but then in the paragraph at the top of page 3, the authors use the symbol g for two different thing in the same paragraph.
* Theorem 7 should be slightly reworded so that it is clear that the first inequality is a condition and not a statement (this is rather obvious in hindsight but for a new reader it is very confusing).
* For denoting multiplication, I would suggest using $\times$ instead of x (e.g. 22x16 etc.)
* Why is there a section 3.2.1 instead of just 3.2?
* I would suggest an overall re-read of the paper to maximize readability.




**Summary Of The Paper:**

The submission proposes new theorems showing the stability of a class of RNNs. Further by combining these RNNs into hierarchical and feedback superstructures, the submission achieve SOTA performance on a number of tasks.

**Summary Of The Review:**

The paper constitutes an evolutionary step in understanding and designing stable RNNs. The theoretical results are novel and noteworthy. Unfortunately the experimental results lack a clear conclusion and at times do not follow best practices (i.e. reporting only the best run out of many).

---

> ### Author Response · Authors · 2021-11-23
> **Thank you for your review, we have substantially reworked the writing of the paper for clarity as well as for consistency, and found your commentary very helpful in doing so.**
>
> Here is a summary of the changes we made as a consequence of your review:
> * Added a reproducibility section (3.2.3) which shows the consistency of our results. This section also provides a control study showing that when our stability condition is not satisfied, network performance suffers. In addition to repeating our SOTA results, we also repeated the experiments with network size (Figure 5). We hope that these additional results together succeed in convincing you that our results are robust.
> * Added columns into our Table 1 which report the mean and minimum of our runs. We note that in the other models we included in this table, the authors reported results from only a single run on each architecture (with the exception of Lipschitz RNN).
> * To maximize readability, we have restructured our results section (3) into clearly demarcated paragraphs and highlighted the main results.
> * More broadly, we have gone through the paper and tried to improve the clarity of the writing. Your comments were particularly helpful here: we replaced ‘x’ with ‘\times’ and corrected the overloading of ‘g’ you pointed out.
>
> As far as the differences between the modularity experiments in our previous sections 3.1 and 3.2, we had focused more on the Sparse Combo Net (since it was higher performing) and ended up not having time to run as many different settings for the SVD Combo Net. So the original modularity experiment in 3.2 actually stops where the graph in 3.1 is still monotonically increasing. We expect modularity to have an inverted U shape for both because complete modularity would mean 1 unit RNN “components”, and plan to do these experiments in the future. The original size experiments are more directly comparable between the two, suggesting that the sparsity and/or static weights in the Sparse Combo Net make it more resilient to issues caused by increasing the number of component RNNs. We plan to further investigate the role of sparsity by regularizing the SVD Combo Net with an L1 weight decay term.
>
> We believe that by addressing your comments we have substantially improved the clarity and results of our paper, so thank you again for the helpful feedback and thorough reading.

---

> > ### Comment · Reviewer_RFwT · 2021-11-23
> > **Reviewer response to initial rebuttal**
> >
> > Thank you for the rebuttal. The modifications have improved the paper and I am increasing my score.
> >
> > A minor but important point: I would like to point out that generally speaking, bad practice in prior work (i.e. running an experiment once) is not a justification for bad practice now. And I would like to reiterate that best performance of an algorithm is not as useful as mean and variance. The best performance is a function of how many times the algorithm is run. Even if a distribution is not fat tailed, if you sample from it long enough, you will draw samples that are significantly larger than the mean + variance. Focusing on the best value will lead to a race of who can run their algorithms more times.
> >
> > Because of this, and in order to improve experimental reporting practice in the community as a whole, I would encourage the authors to focus on mean + variance and then also make their numbers with the highest mean bold. Then on the side they can also report the best value they achieved while highlighting how many runs it took to get this number. (I agree this would not be an apples to apples comparison to prior work since they only report one run, but this will be implicitly clear since prior work has no +/- error values.)

---

> > > ### Author Response · Authors · 2021-11-28
> > > **We definitely agree with you on the problems with reporting only best results.**
> > >
> > > We had tried to improve upon existing practices in our original submission by including repetition info in the main text, as well as detailing our results from across hyperparameter tuning and all experiment trials in the appendix. Putting repetition information in Table 1 was a substantial improvement on that, so thank you again for the suggestion! We can't update the PDF during this period, but we will further tweak the table before final submission to ensure mean/variance and number of repetitions are highlighted (flip column order, bold mean number where appropriate, update caption). We will also add the mean and variance to the paragraph where we state our key results. In order to maintain a 1 to 1 comparison we do plan to keep the best result column in the table though.
> > >
> > > We have continued to run repetitions of the best network settings on the sequential CIFAR10 task since submitting our revision, so we will have a larger sample to report final mean and variance from. The current numbers are n=7 with mean 64.64 and variance 0.447. We plan to run a few more repetitions and then update the manuscript accordingly.
> > >
> > > Thank you very much for helping us to improve our paper and subsequently reevaluating the score!

---

### Official Review · Reviewer_V8Ay · 2021-11-01

**Correctness:** 4
**Technical Novelty And Significance:** 3
**Empirical Novelty And Significance:** 2
**Recommendation:** 8
**Confidence:** 3

**Main Review:**

Strengths:
- I thought that the empirical results were rather convincing for what is primarily a theoretical contribution. The authors first thoroughly investigate various permutations of their modular sparse combination network framework (# RNNs vs size of each using absolute value weight constraints) and do another investigation of their alternative SVD weight constraint network (which doesn’t perform as well or train as quickly). Most importantly, they then show that they can best SOTA algorithms on some of the common (albeit easier) benchmarks in the field, even under (and perhaps because of) these constraints.
- The theoretical contribution is quite powerful. There has been a lot of recent work in networks with many individual recurrent components, such as the aforementioned AlphaGo or the more general recurrent independent mechanisms (RIMs) framework, but for the most part, they rely on intuitive explanations and empirical results over theoretical guarantees. Clearly specialized RNN modules can be quite powerful, but RNNs are notoriously unstable and difficult to learn, and learning such models end-to-end is tricky. If we can apply these constraint conditions and still achieve good performance (which seems like it could be realistic, particularly in the absolute value constraint case), then we can develop sets of useful modules and mix-and-match to the task in question. This paper doesn’t answer all of the intermediate questions, but the stability analysis is a key step.
- The proofs in the appendix are well-done and easy-to-follow, given a sufficient math background.

Weaknesses:
- This paper is very dense and difficult to follow. It took me a few reads to really understand the value of network stability and how it’s achieved in this case. The appendix is a mandatory read as are some of the references. None of the use cases are particularly intuitive. I think I would have liked to see a graphical representation of the sparse combo network (rather than the weight matrices in Figure 2), some pseudocode for the algorithms (tossed in the Appendix), and maybe an example case of an unstable network assembly diverging. I also feel like my familiarity with AlphaGo and other methods gave me more of an insight into how this would help in practice than the actual paper did.
- As much as I liked the empirical results that were provided, they’re all of a kind: sequential image prediction. I would have liked to see at least one application in a different domain (NLP, RL, continuous control, etc).


**Summary Of The Paper:**

This paper is primarily a theoretical contribution to the construction of assemblies of recurrent neural networks. We know that combinations of learned modular components can be powerful and far more tractable than learning bespoke models from scratch, particularly in applied domains (e.g. AlphaGo). Yet so far, we have no theoretical guarantees that these combinations will actually remain stable. This paper develops the theory behind provably-stable combinations of RNNs using weight constraints and feedback mechanisms. Then, using fixed RNNs generated according to these constraints (leaving the connections between them as antisymmetric learnable parameters), the authors show that their sparse combination network is able to achieve SOTA performance on sequential image classification benchmarks with far fewer learned parameters and the previous stability guarantee.

**Summary Of The Review:**

Overall, I would accept this paper. Although it was difficult to follow and required a lot of consultation with the literature, I do ultimately think that this is a direction that DL algorithms are going in and that the theoretical and practical results from this work could be quite powerful. To make the paper better, I would like to see some results in a different domain and more effort towards improving the readability. Too often, valuable theoretical works go underutilized because they’re difficult to understand or don’t seem relevant to the empiricists and engineers who could build on them.

---

> ### Author Response · Authors · 2021-11-23
> **Thank you for your review, we really appreciate your thorough reading.**
>
> We’ve submitted a revised manuscript to improve readability, particularly of the experiments section. At your suggestion, we included a basic graphical representation of the network in the main text, and some more detailed information on the architecture setup in the appendix, including code. We also worked to better connect the theoretical and experimental results in our writing, and have included some explanations at more levels of abstraction to hopefully make the paper approachable for a wider audience. We are planning to use this architecture in some RL experiments in the future, as this is the application we are most excited about. Thank you again for your positive comments and thorough reading.

---

> > ### Comment · Reviewer_V8Ay · 2021-11-25
> > **Follow-up**
> >
> > I appreciate the work that's gone into making this more legible! I realize you're a bit cramped for space, but the captions for the graphs (immediately underneath the boxes) aren't really readable without zooming in quite a bit. I think those could honestly be moved to the main figure caption instead anyway (separate by (a), (b)). Otherwise, the clarity has improved quite a bit. I also appreciate the addition of more information in the empirical results.
> >
> > Like the other reviewers, I agree that I'd still prefer to see mean/variance. The previous works didn't provide this and that it's not your job to reproduce this kind of data for their methods, but if you can do it for your own methods, then that will both demonstrate your methods' robustness and also allow future work to build on a stronger foundation.

---

> > > ### Author Response · Authors · 2021-11-28
> > > **Thank you again for your suggestions!**
> > >
> > > Thank you for pointing out the issue with Figure 2, we can't upload a new PDF during this period but we've improved the readability in our local copy so we can update it when possible.
> > >
> > > Since submitting the revised paper we have continued to run some repetitions of the best network settings on the sequential CIFAR10 task. We have now reached n=7 with mean 64.64 and variance 0.447 (min 63.73, max 65.72). We currently have another repetition in progress, and plan to reach n=10 before final publication. When we can update the manuscript we will add the current numbers to the main text, making sure the mean and variance are highlighted at that time.

---

> > > > ### Comment · Reviewer_V8Ay · 2021-11-29
> > > > **Confirmation**
> > > >
> > > > I appreciate it, the paper has definitely improved and with those changes would improve further. I'm not comfortable raising to a 10 (next level up), but I'm firmly positive on the paper and my original score

---

### Official Review · Reviewer_hNbM · 2021-11-03

**Correctness:** 4
**Technical Novelty And Significance:** 3
**Empirical Novelty And Significance:** 3
**Recommendation:** 6
**Confidence:** 2

**Main Review:**

Strength
How to assemble a network of RNNs is an interesting problem. The theorems on contraction properties are helpful to people thinking about provably stable RNNs.


Weaknesses
(1)	Section 3.2.1 is pretty dry to read. Reporting results from many individual AxB networks seem unnecessarily.

(2)	The authors showed performance comparison with other types of networks in Table 1. I think it would be quite informative to show performance of networks where everything is kept the same, except that the RNNs are no longer provably stable.

(3)	It would also be good to know what happens if all connection weights are trained, not just the connections between modules. Does the performance actually decrease despite having more parameters?

(4)	The provably stable part is kind of separated from the training modular network part. How closely are they related? Is having stable RNN modules particularly important for sparsely connected modular networks?



**Summary Of The Paper:**

The authors studied contraction properties of continuous-time recurrent neural networks. They further showed that a network of provably stable RNNs (net of nets) can be trained to reach competitive performance on several benchmarks, including sequential CIFAR10, even when only connections between modules are trained.

**Summary Of The Review:**

Overall, this is an interesting paper that takes a less common approach to RNNs: provable stability and net-of-nets. The results are at places more difficult to read, but overall it is clear.

I want to add that I cannot evaluate whether the mathematical derivations are correct.

---

> ### Author Response · Authors · 2021-11-23
> **Thank you for your review, we have substantially reworked the writing of the paper for clarity, and found your commentary very helpful in doing so.**
>
> We have tried to address and correct all the weaknesses in our paper you listed. In the order you listed them:
> * As you suggested, we ran experiments on the SVD Combo Net and Sparse Combo Net which held the stability conditions on the individual RNNs fixed, but left the connections between subnetworks unconstrained. Therefore the overall RNN was no longer provably stable. We found that doing this decreased performance on permuted seqMNIST by ~60% on the Sparse Combo Net and 0.6% on the SVD Combo Net. We have written these results into our paper (section 3.2.3).
> * Our first draft did not clearly communicate that SVD Combo Net does in fact undergo training of all its weights, and not just the weights between subnetworks. The Sparse Combo Net undergoes training on only the weights between subnetworks. We have rewritten the paper to better convey this information. Additionally, we have rerun our experiments on permuted seqMNIST using both these networks to understand the effect of network size on performance--the results may be seen in Figure 5.
> * We have added a new section 2.2 which more explicitly states the ‘network of networks’ model. We hope that this section provides a clear link between the theorems for individual networks presented in section 2.1 and the experimental results discussed in section 3. In addition to this section, we also included the Pytorch code for the network of networks, which we hope aids in establishing the connection between the individual RNN stability results of section 2.1 and the network of network stability results in section 2.2.
>
> Thank you again for the helpful feedback and thorough reading.

---

> > ### Comment · Reviewer_hNbM · 2021-11-23
> > **Response**
> >
> > I have read the other reviews and the authors' response. I have no further questions. Thank you!

---

### Official Review · Reviewer_q4Si · 2021-11-03

**Correctness:** 4
**Technical Novelty And Significance:** 2
**Empirical Novelty And Significance:** 2
**Recommendation:** 5
**Confidence:** 4

**Main Review:**

The theoretical results seem interesting, although not very surprising. However, I think there is disconnection between the theoretical results and the proposed model: not all theorems are relevant to the proposed model.

For the proposed model (Section 3.1), I might have missed something, but I fail to fully understand the model; the presentation could be improved to help the readers. For example, the mentioning of "subnetworks" at the beginning of Section 3.1 is not defined/explained. It's unclear to me how the subnetworks are combined. I can only infer from the "recursive construction" in the title and Figure 2 that the resulting weight matrix is a block matrix.

Furthermore, the idea of parametrizing orthogonal weight matrices by exponentiating skew-symmetric matrices is not novel and has been explored in expRNN [1].

The writing of the introduction section could also be improved. Instead of discussing AlphaGo and modules in evolution, the reader might benefit from a more thorough literature review of the RNN trainability and long-term dependence.

[1] Lezcano-Casado, Mario, and David Martınez-Rubio. "Cheap orthogonal constraints in neural networks: A simple parametrization of the orthogonal and unitary group." International Conference on Machine Learning. PMLR, 2019.

**Summary Of The Paper:**

In the paper, the authors study stable architectures for RNNs. On the theoretical side, the authors present a series of conditions such that a weight matrix of an RNN is contractive. On the modeling side, the authors propose RNN architectures that have contractive weight matrices. The proposed methods are evaluated on benchmark datasets including sequential MNIST, permuted MNIST, and sequential CIFAR-10.


**Summary Of The Review:**

The theoretical results in the paper seem interesting. However, the presentation of the proposed model is not clear, and the model itself does not seem novel. Overall, I think the paper needs improvement to meet the acceptance threshold.

---

> ### Author Response · Authors · 2021-11-23
> **Thank you for your review, we have substantially reworked the writing of the paper for clarity, and found your commentary very helpful in doing so.**
>
> One of the points of clarification we focused on in our revisions was the connection between the theoretical results and the experimental architecture, as it was not well communicated previously. Towards the goal of improving this weakness in our paper, we have added in a new section (2.2) which explicitly describes the ‘network of network’ model we use, as well as a new figure (Fig 2). We have also added code for the models used into the appendix to help aid in clarity, and have overhauled writing in the experiments section to provide a better verbal description of our methodology and results.
>
> Another point of clarification is the relation to previous work, in particular work parameterizing RNNs using orthogonal weights to avoid an exploding/vanishing gradient. We have added a paragraph to the introduction (third paragraph in 1.1) which compares our work to this work. To briefly state the point here, the work you cited--as well as essentially all other work on stability in RNNs--seeks stability conditions for individual RNNs, not embedded into any larger subnetwork. By contrast our work deals with stability in ‘networks of networks’. We show that certain ‘local’ stability conditions for individual RNNs, taken together with certain rules for connecting these RNNs into a network of networks, preserve stability during training. One of these local conditions--Theorem 5--can be parameterized by using what is essentially a singular value decomposition, which requires the use of two orthogonal matrices. We parameterize the orthogonal matrices by using the exponential of a skew-symmetric paper as done in the work you cited. Therefore we should have cited this work and have now included it in our paper.
>
> To comment on the ‘surprisal’ value of our work: we agree our stability conditions are quite intuitive. However, as we discuss in the paper, not all intuitive conditions are easy to prove or even true. Prior works have cited ‘theorems’ that are in fact not actually true, as we show in the paper (section 2.3). Thus rigorous proofs are an important contribution to the literature. Moreover, modular networks are of inherent interest to many researchers in biology and machine learning, and so the applicability of our conditions to such networks is a major advantage, as stability conditions for non-linear systems in general do not necessarily carry any guarantees related to stability of combinations of systems. Thus the novelty of the experimental work is in the combination of stability constraints with modularity principles. Our architecture outperforms existing state of the art for provably stable networks, which we believe is a direct result of our “network of networks” approach.
>
> As part of our rewriting process, we took great care to distinguish between the two architectures we experimented with throughout our methods and mathematical results sections. In conjunction with the improved description of these architectures, we believe our new submission will address many of your concerns. Thank you again for your comments which we believe help to substantially improve the clarity of our paper.

---

> > ### Comment · Reviewer_q4Si · 2021-11-30
> > **Response**
> >
> > I appreciate the effort the author put into improving the paper. The added sections (1.1 and 2.2) definitely make the paper more readable. However, I think there is still the disconnection between the theory and the proposed method. Also I'm still concerned about the insufficient novelty. Therefore, I have raised my score to 5.

---

> > > ### Author Response · Authors · 2021-11-30
> > > **Novelty**
> > >
> > > Thank you very much for increasing your score, we appreciate your feedback.
> > >
> > > We would like to emphasize that our best performing architecture and the primary focus of the results section does not use the parameterization mentioned in [1]. In our Sparse Combo Net, we utilize sparse initialization to achieve stability and then fix the internal subnetwork weights, training only the between-module weights while enforcing contraction combination conditions. Thus the model is a significant departure from existing literature, in addition to the novelty of the theoretical implications for stability of modular networks.

---

> ### Author Response · Authors · 2021-11-29
> **Follow-up**
>
> Hello, we thank you again for your helpful initial comments. We hope that in light of our substantial revisions, as well as our comments to you and the other reviewers, you will consider reevaluating your score. Please let us know if you have any further questions, and thank you again for your time!

---

### Author Response · Authors · 2021-11-28
**Summary of revisions**

Thank you again to all reviewers for your helpful suggestions. As we are nearing the end of the discussion period, we wanted to recap the major changes made in our updated manuscript.

The clarity of the paper, in particular the empirical portion, was a major point of concern across reviews. To address this we did a substantial rewrite, including the following changes:
* Added a section (2.2) to provide theoretical details on the use of our single-RNN stability conditions in constructing a stable 'network of networks'.
* Added a diagram (Figure 2) summarizing our experimental architectures. The appendix also now includes code snapshots with the network class definitions and a more detailed schematic.
* Restructured the experimental section (3) to better describe the architectures, highlight the key takeaways, and organize results details by category.
* Included more background (section 1.1) on prior results in RNN stability research and how our work sets itself apart.

The other concern expressed by multiple reviewers was related to verifying the robustness of the results. The updates we made to address this include:
* Repeated trials of our best settings on the sequential CIFAR10 task (n=3 in the revised PDF, up to n=7 currently).
* Included information on repeatability in Table 1, which summarizes results on sequential image classification tasks across relevant papers in comparison to our results with the Sparse Combo Net.
* Reran all trials demonstrating the effect of network size on performance, for both our architectures.
* Ran control trials for both our architectures where the connections between subnetworks were trained with no constraints.
* Made sure repeatability and control results were thoroughly described in a clearly demarcated subsection (3.2.3).
* Included the Jupyter Notebook used in training and evaluation of the Sparse Combo Net as a supplemental upload.

We believe these changes have greatly benefited our paper, making clear the novelty of our stability conditions and in particular their use for constructing stable 'networks of networks', as well as strengthening our arguments for the experimental utility of our approach. We are pleased to see that thus far the reviewers agree we have improved upon the paper's weaknesses, adding to the contributions of our original theoretical results. Thank you very much for all the work you put into bettering this paper!

---

> ### Public Comment · ~Michaela_M_Ennis1 · 2022-02-07
> **.**
>
> (replied to this comment accidentally, reposted response correctly under "Paper Decision")

---

### Decision · Program_Chairs · 2022-01-20

**Decision:**

Reject

**Comment:**

In the context of recurrent neural networks, the motivation of the paper is to explore the "space" between fully trained models and almost not trained models, e.g. echo state networks, using a formal approach. In fact, a modular approach has proven to be very successful in many practical applications, and in addition brain seems to adopt this strategy as well. The addressed theoretical issue is stability of the network (i.e., the network implements a contraction map.) Specifically, it is assumed that a network is composed of a set of subnetworks that meet by construction some stability condition, and the problem is to design a mixing weight matrix, interconnecting the latent spaces of the subnetworks, able to give stability guarantees during and after training. Some novel stability conditions are proposed as well as two different approaches to design a successful mixing weight matrix. The original submitted paper was not easy to read, and after revision major problems with presentation have been resolved, although the current version looks more like an ordered collection of results/statements than a smooth and integrated flow of discourse. The revision has also addressed some concerns by reviewers on the role of size and sparsity of the modules, as well as the sensitivity of the stabilization condition on the mixing weight matrix has been experimentally assessed, obtaining interesting results. Overall the paper reports interesting results, however the novelty of the contribution seems to be a bit weak, e.g. stability conditions on recurrent networks (although different from the reported ones) were already presented in literature. Also the idea of exploiting, in one of the proposed models,  the fact that the matrix exponential of a skew-symmetric matrix is orthogonal to maintain the convergence condition during training, is not novel. Moreover, the experimental assessment does not provide a direct comparison, under the same architectural/learning setting, of the novel stability results versus the ones already presented in literature. Empirical results are obtained on simple tasks (using datasets with sequences of identical length), and relatively small networks, which limits a bit the scope of the assessment, as well as it is not clear if the observed improvements (where obtained) are statistically significant (especially when compared with results obtained by networks with the same order of parameters.) The quality of the assessment would increase significantly by considering datasets with sequences of different lengths, and involving more challenging tasks that do require larger networks.

---

> ### Public Comment · ~Michaela_M_Ennis1 · 2022-02-07
> **Clarifications**
>
> Thank you for all you do to make ICLR happen! We just wanted to clarify a few major points for the sake of those that may glance at this OpenReview in the future. Particularly because our overall scores indicated likely acceptance, we will be submitting to another conference and would not want any potential misconceptions to propagate.
>
> Regarding the novelty of our paper, we prove stability conditions for modular 'networks of networks', something not previously addressed in the RNN stability literature. This in itself is a novel and substantial contribution, as supported by the majority of our reviewers. It is mentioned within the first few sentences of review V8Ay, so we were surprised to see that the meta review did not even acknowledge this point.
>
> Additionally, we do prove new stability conditions for the individual RNN model that has previously been studied, as you mention. Existing conditions for this model are quite limited, and so we are confused why further research on the topic would be assumed 'not novel' inherently. In fact, as part of our paper we provide counterexamples to a few of these supposed conditions, further calling into question their use to dismiss the novelty of our work.
>
> The parameterization we use to maintain stability of *components of* SVD Combo Net throughout training is indeed not novel, but this was never claimed to be part of the paper's novelty - rather a tool used in implementing one of our architectures, and not even the architecture of primary focus. We hope that future discussion of novelty can center around the parts of our work that we do claim to be novel. Most importantly, the study of stability conditions in *combination* networks, but also the new conditions presented for stability of the individual nonlinear RNN model with static weights.
>
> As far as the experiments, all the models we compare to underwent their own hyperparameter tuning process, and performance results across hyperparameters for these models were unable to be found (even in the associated appendices). Thus we did as direct a comparison as we could by performing our own hyperparameter tuning process on the same benchmark tasks, while keeping the number of trainable parameters comparable to the prior SOTA for stable RNNs. We also took great care to report results across every trial we ran with Sparse Combo Net in our appendix.
>
> Unfortunately, very few of the cited papers provide repeatability information for the best settings of their architectures either, and none provide this information for the more difficult sequential CIFAR10 task. We would have loved to do a deeper comparison with these works, but we did not have the information necessary to do so. We report the results of all of our own repetitions to ensure that future studies can do such an analysis in comparison to our work.
>
> We agree that many of the additional experiments you've proposed are of great interest, and this is a future direction we are actively exploring as a follow-up project. However, we strongly believe that our paper's contributions already justify its publication in a venue such as ICLR, due to the important theoretical groundwork they lay. This belief is also consistent with our reviews, and it was even stated that "the empirical results were rather convincing for what is primarily a theoretical contribution".
>
> Ultimately, we would like to thank the reviewers again for their help in making substantial improvements to our paper. While we are disappointed by the final decision, submitting to ICLR has been worthwhile because of these interactions.